# Carbon Anode in Carbon History

**DOI:** 10.3390/molecules25214996

**Published:** 2020-10-28

**Authors:** César A. C. Sequeira

**Affiliations:** CeFEMA, Instituto Superior Técnico, Universidade de Lisboa, 1049-001 Lisbon, Portugal; cesarsequeira@tecnico.ulisboa.pt

**Keywords:** carbon materials, catalysis, lithium ion batteries, carbon fuel cells, electrochemical capacitors, neurochemical monitoring

## Abstract

This study examines how the several major industries, associated with a carbon artifact production, essentially belong to one, closely knit family. The common parents are the geological fossils called petroleum and coal. The study also reviews the major developments in carbon nanotechnology and electrocatalysis over the last 30 years or so. In this context, the development of various carbon materials with size, dopants, shape, and structure designed to achieve high catalytic electroactivity is reported, and among them recent carbon electrodes with many important features are presented together with their relevant applications in chemical technology, neurochemical monitoring, electrode kinetics, direct carbon fuel cells, lithium ion batteries, electrochemical capacitors, and supercapattery.

## 1. Introduction

At the start of things, that is more than 15 Gy ago, all the matter and energy that we can observe was concentraded in a volume element about the size of a small coin (~100 mm^3^). Later, within resultant stars, at temperatures of about 10^15^ K, hydrogen atoms were stripped of their nuclei and fused to form helium nuclei. As stars cooled, collision of helium nuclei led to beryllium, of fleeting stability but of sufficient stability to allow a further collision with a helium nucleus, to give us carbon. Also, a continuing collision of carbon with a helium nucleus gave us oxygen; and so our story has started.

Some 5 Gy ago, from of the cosmic dust out there, an event occurred leading to the formation of the solar system, with the sun, planets, and moons. Only one planet (earth) was of the correct size and at an optimal distance from the sun to create and maintain oceans and an atmosphere. The atmosphere was first made up by volcanic activity, of carbon dioxide and water vapour, two greenhouse gases. Eventually, nitrogen was added to the atmosphere. The oceans were created and carbon dioxide was established within the carbon cycle with formation of carbonates. These greenhouse gases kept the atmospheric and oceanic temperatures compatible with chemical reactions leading to molecules evolution, i.e., the creation of life. The oceanic plant forms took out the carbon dioxide from the atmosphere and replace it with oxygen (photosynthesis). These early plants became entrapped within the rocks and their remains are now identified as kerogens, and petroleum deposits and methane reservoirs. As the continents of earth moved over its surface the great rainforests were established, which provided the organic material that, after being chemically degraded and fossilised, is now recognizable as coal. Nature in the process of maturation of kerogen and coal, had, as the end product, graphitic material, which is the basis of all carbon forms, with the exception of crystalline diamond. About 15,000 years ago, petroleum and coal appeared making life very easy for modern man. The waste products of the petroleum industry (the distillation of the barrel), and the waste products of the coal carbonization industry (the manufacture of metallurgical coke) as coal-tar and coal-tar pitch, are used to create the carbon artefacts (matrix and binder) of the carbon electrode industry [1].

The development of the several segments of the carbon industry can be seen to be quite closely related. They did not develop independently of each other, really. The years after WW-II saw extremely rapid advances in mass production of consumer goods and in mass transportation. A starting point for our story could be with Henry Ford and his motor cars (Figure 1).

The automobile had, at least, two basic requirements; it needed metal for its production and petrol (gasoline) for its mobility. Iron metal production soared, coking plants were built to feed the blast furnaces, and coal-tar and coal-tar pitch were indeed plentiful (a glut). Petroleum companies were growing in number and size and gasoline availability soared. However, not all of the barrel could be converted to gasoline or jet fuels, or fuel oils. The remnants of the barrel, asphaltenes, petroleum pitch residue, had no comercial value and presented a disposable problem. Land fill was out of the question and use as a fuel was impractical (unlike coal-tar pitch). The delayed coker was developed to take care of growth in amounts of disposable pitch and the words ‘delayed coke’ were heard more frequently. Landfill and use as fuel were more pratical with a solid.

During the first half of past century, it had become obvious that the route to aluminium production was via the Hall-Héroult cell, i.e., the electrochemical reduction of alumina, by carbon, in a molten bath of cryolite. Developments of the carbon anode had pointed the way to the use of a coke bonded with coal-tar pitch. At the same time as the aluminium industry was expanding, the petroleum and steel making industries were providing the necessary ingredientes of anode manufacture. Of course, refinements in the quality of residues going to the delayed coker were necessary (to make regular and needle coke, as distinct from shot coke), and more stringent specifications were applied to coal-tar pitch quality. However, one situation has not changed over all of these years, which is that the aluminium industry has to cope with the problems of quality control of its essential supplies, considered by the suppliers as waste materials. There is an additional complication in that petroleum and coal resources are changing with exploitation and hence continuous quality control of coke and pitch for the anode is a necessity [2,3,4,5].

In terms of the history of the carbon industries, carbon blacks impinge into our story. Although dominantly associated with printing inks, carbon blacks are an essential ingredient of the automobile tyre. Aircraft use braking systems of carbon composites made up of carbon fiber matrices bonded with carbon from coal-tar pitch. Also, the steel industry makes its steels in the furnace heated using the grafite electrode, made from premium quality delayed coke (needle coke) and coal-tar pitch. In the present century, sophisticated carbon electrodes have been manufactured for many applications, namely in the area of electrochemical energy devices [6,7,8].

Light metal meetings and many other meetings involving several industries using carbon electrodes continue to show that there are many factors that need to be considered and improved to obtain efficient anode electrodes. Structure and purity of delayed cokes, permeability, the role of catalytic impurities, wettability by the pitch, the physical properties of the pitch (e.g., the relevance of glass transition temperatures), and the role of QI material in pitch all continue to dominate the discussions. Restricting to the aluminium electrowinning, we really do not understand very well what happens within the green anode when we pyrolyse and bake it, namely we do not know exactly how to moderate optimum relationships between coke particle (shape and size), butt particle, and mixing extent with coke particles, and the shape and size of the binder coke bridges [9]. During pyrolysis and baking, I suspect that we have little true idea of the interactions that are occurring between the pitch and, later, the mesophase derived from pitch with the particulate components of the green paste. Summarizing, there is still a great future for the light metals meetings and related meetings on carbon materials.

At this point, it seems that it is time to stress that not all days are black in the carbon world. Looking back at the carbon highligts, we clearly find some areas deserving attention. It is the case of the aluminium production in large alumina refineries, using carbon anodes of high quality, which depends on the characteristics of coke filler, coal tar pitch binder, and anode scrap, among others [5]. However, then we can see the development of synthetic diamonds by the GE high-pressure catalytic process initiated in 1941 and leading to the first commercially successful synthesis on december 1954 [10,11]. Much later the diamond and diamondlike films appeared, using low temperature and low pressure, truly defiant of all the laws of thermodynamics and phase diagrams [12,13]. The carbon fibers, emerging first from PAN, and later from pitch, are other excellent carbon materials whose development led to the carbon fiber reinforced plastic (CFRP) and other composite products, which have several uses in aerospace and non-aerospace structures, as well as in non-structural applications (thermal insulation, electrodes for batteries and supercapacitors, hydrogen gas storage, etc.) [14,15,16,17]. One of the most exciting events of all has been the explanation for the formation of anisotropic, graphitizable carbons via the intermediate phase of mesophase, that nematic, aromatic, discotic liquid crystal system [18]. However, probably the most intriguing discovery has been the fullerene systems and the nanotubes, including curved crystals, inorganic fullerenes and nanorods, hybrids of carbon nanotubes and graphene, carbon anions and spheroidal carbon particles, and other related nanosctructures that are capturing the imagination of physicists, chemists, materials scientists, and nanotechnologists alike [19,20,21,22]. These new discoveries and developments had an impact that extends well beyond the confines of academic research and worked the beginning of a new era in carbon science and technology [23,24,25,26]. In this century, the progress is still slow, but applications begin to appear, and future prospects are enormous. Moreover, the field of carbon electrochemistry has experienced a robust development over the last decades with the emergence of the multidimensional carbon materials cited above [26,27].

In general, carbon-based electrodes are characterized by low cost production, high surface areas, a wide working potential window in many media, high electrocatalytic activities for different redox-active biochemical systems, and chemical inertness. Moreover, their surface chemistry enables the functionalization of these carbon platforms via strong covalent or noncovalent methods with surface modifiers, which improves their electrochemical performance [28,29]. The research interest on carbon for electrocatalysis is also stimulated by the need to develop efficient electrodes for energy utilization (from fuel cells to batteries, photoanodes, and solar cells). In fact, to meet the demanding expectation for more sustainable and efficient conversion and storage of energy it is necessary to give proper attention to the conductive properties of (some) carbon materials and the possibility of fine tuning their nanostructures [30,31]. The recent recognition that carbon materials used in electrochemical devices exhibit catalytic behavior in addition to electrochemical properties has moved many research groups working in catalysis to the electrochemical field bringing back new expertise on catalysis.

For these and other reasons, the aplication of advanced carbon-based materials was fast growing over the last decade. As a matter of fact, there was an exponential increase in the field of carbon and catalysis, particularly in nano and electrocatalytic aspects during the last decade [32,33,34,35,36,37]. This was the motivation to contribute a communication to the Symposium ‘Carbocat VIII’ directed particularly at the carbon anode and its traditional and new possibilities for the 21st century. More specifically, this article examines, briefly, how the several major industries, associated with carbon artifact production, essentially belong to one, closely knit family, whose common parents are the geological fossils called petroleum and coal, and, also, attempts to review some important applications of carbon electrodes, with a major focus on anode electrocatalysts developed over the last 30 years or so. The next section begins with a brief account of structure in carbons and carbon forms, followed by catalysis of carbon oxidation reactions, nanotechnology, and carbon electrocatalysis. The following sections deal with carbon anodes for the aluminium smelter electrolysis, and then summarize, briefly, recent advances of carbon materials and nanomaterials as anodes in newer electrochemical energy technologies.

## 2. Structure in Carbons and Carbon Forms

The element carbon has an atomic weight of 12.011 and is element number 6. Three isotopes are known: 12 C, 13 C, and 14 C. The natural abundance of the stable isotopes is: 12 C–98.90%; 13 C–1.10%. The radioactive isotope 14 C, which is generated in the upper atmosphere by neutron bombardment of nitrogen (14 N + n = 14 C + 1 H), has a half-life of 5730 years. As well as being used for dating archaeological artifacts, 14 C is useful as a tracer in the study of organic reactions. With its magnetic moment (spin ½), 13 C is ideal as a probe for NMR studies.

Because of its large abundance and combining power, 12 C is used as the reference definition for atomic mass, being defined as having the Relative Atomic Mass of 12 exactly. All other atomic and molecular masses are now based upon this definition.

For most carbon science requirements, the isotopic composition of the carbon is irrelevant as the properties are governed by the electronic configuration. The orbital arrangement of electrons, where the superscript indicates the number of electrons in the corresponding sublevel, is 1 s^2^, 2 s^2^, 2 p^2^. Seven isotopes of carbon are known: C10, C11, C12, C13, C14, C15, and C16, with the isotopes 12 and 13 being stable, and the others are radioactive. It is estimated that more than 1.5 million carbon compounds are described in the chemical literature, and chemists synthesize many new ones each year. Much of the diversity and complexity of organic forms is due to the capacity of carbon atoms for uniting with each other (they display catenation) in various chain and ring structures and three-dimensional conformations, as well as for linking with other atoms. Many of these structures are the carbons, which are the subject of this paper. Carbon (mainly in the combined state) is widely distributed in the earth’s crust, though it comprises only about 0.2% of the outer 10 miles. Carbon dioxide, which constitutes approximately 0.03% by volume of the atmosphere, is found also in all natural waters. Carbon is a constituent of coal, petroleum, and natural gas, and of many minerals.

The formation of σ- and π- bonds between carbon atoms and with other atoms (e.g., N, O, etc.) leads to the possibility of extensive and complex structures manifest in a whole branch of chemistry (Organic Chemistry) devoted to carbon compounds. The stability of carbon bonds and, in particular, the multiple bonding avaible through π- bonds is a principal feature of Carbon Science [38].

Carbon is an element with a unique ability to bond with itself principally via sp^2^ (graphite-like) and sp^3^ (diamond-like) hybridization. The hybrid orbitals can then be assumed to link with compatible orbitals on other atoms to form σ- bonds while the p-orbitals are free to form π- bonds. The resultant structures have an immense variety of possibilities but for most of the materials dealt with in carbon science they can be considered as composed of mainly graphitic subunits, with more or less structural order, linked together by less ordered regions.

Only three of the many forms of carbon can be definitely characterized: Diamond, graphite, and black carbon, all stable at ordinary temperatures and insoluble in common solvents. Diamond and graphite are crystalline, black carbon is more or less amorphous, depending on the mode of preparation, and there are many varieties. A brief description of these generic carbon forms is given first.

At ambiente temperatures and pressures, graphite is the most thermodynamically stable of the two regularly ordered allotropes, graphite and diamond, of carbon: C (Diamond)→C (graphite)   ΔH = −2.1 kJ/mol(1)

However, from a kinetic point of view, the change is extremely slow at room temperature (rapid at about 1900 K) because of the large number of bonds that would need to be broken in the process.

Within the diamond lattice each carbon atom is surrounded by four others in the form of a regular tetrahedron; this arrangement confirms the tetravalency of carbon and the postulated directionality of its valence bonds (Figure 2). Pure diamonds are colourless and transparent, but they are frequently coloured red, blue, green, or yellow because of small amounts of impurities. Diamond is the hardest substance found in nature, with a value of 10 on the Mohs scale. Thermodynamically, it is more stable than graphite at pressures >60 GPa at room temperature principally due to its higher density (3.51 g·cm^−3^) compared with that of graphite (2.25 g·cm^−3^). Within the diamond lattice, the bonding electrons are fixed between atoms so that electrical conductivity is very small.

Graphite is widely distributed in nature, being found as soft, gray-black, shiny-leaflets, and unlike diamond is a good conductor of heat and electricity. Its structure consists of connected planar hexagonal rings linked vertically to similar planar rings, thus producing the characteristic layer arrangement of graphite. In graphite, there is both σ- and π- bonding holding the atoms in hexagonal two-dimensional networks (Figure 3).

Black carbon or amorphous carbon includes coke, lampblack, carbon black or gas black, gas carbon or retort carbon, soot, and the charcoals, of which there are many varieties, each depending upon the process employed in the manufacture and upon the starting material. Actually, X-ray examination of various amorphous carbons has shown them to be more or less crystalline, resembling graphite in structure; but there are many intermediate stages between amorphous carbon and graphite. All varieties are more or less readily attacked by the strong oxidizing agents that react with graphite. Carbon black is used extensively in the manufacture of automobile tires and printer’s ink. The charcoals (sugar charcoal, wood charcoal, animal charcoal, or bone charcoal, etc.) are very porous, and consequently, their specific gravity is apparently only about 0.25, but when the air is pumped out of the pores this becomes 1.4–1.9. Charcoal is hard and brittle and is a poor conductor of electricity. Like soot, it is dead black and without luster. An important variety of black carbon is activated carbon, made by heating inactive carbon in steam (or other gases). It has a great capacity for adsorbing dissolved substances and gases. This property makes activated carbon valuable as a decolourizing agent in sugar refining, and as an agent to remove gases from contaminated air. It is the principal component in gas masks.

Carbon forms and their definitions and key properties, have been published periodically in the Journal Carbon by the International Committee for Characterization and Terminology of Carbon, the most important being the following: Graphitic carbons, graphite, natural graphite, synthetic graphite, non-graphitic carbons, non-graphitizable carbons, graphitizable carbons, coal-tar pitch, petroleum pitch, coke, green coke, calcined coke, petroleum coke, coal derived pitch coke, metallurgical coke, delayed coke, sponge coke, needle coke, coals, char, carbon fibers, charcoal, carbon blacks, activated carbons, mesocarbon microbeads, diamond-like films, graphitic composites, carbon electrodes, and carbon/carbon composites. During the development of these forms, several formation processes are largely applied, being appropriate to define them. It is the case of carbonization, graphitization, and coalification. Carbonization is a process of formation of material with increasing carbon content from organic material, usually by pyrolysis, ending with an almost pure carbon residue at temperatures up to 1600 K. In other words, is the common pyrolysis used by most of the solid carbons to derive from organic precursors. Graphitization is a solid-state transformation of thermodynamically unstable non-graphitic carbon into graphite by thermal activation. In other words, it is the conversion of many non-graphitic carbons into graphitic carbons by heat treatment to above 2500 K. Also, coalification is a geological and chemical process of dehydrogenation, deoxygenation, and condensation, which occurs in the earth’s crust by gradual transformation at moderate temperatures (±500 K) and high pressures [39,40,41,42,43,44,45]. Apart from the carbon forms already described, there are a few more requiring consideration, namely the mesocarbon microbeads, the diamond-like films, the graphite composites, and the carbon electrodes. Mesocarbon microbeads is the term introduced by Honda and Yamada [46] to describe the mesophase spheres generated on heat-treating pitches and separated by solent extraction or other means. Work by Auguie et al. [47] has shown them to have the classical Brooks and Taylor structure. The control of the size and the morphology is an expanding field of study. Mesocarbon microbeads have many potential applications in mechanical carbons, as filters and as adsorbates [48]. Diamond-like films, the development of which has taken place over the past few years, have provided much interest in structures which have little graphitic character. These films are usually produced by arc discharges in hydrocarbon gases in the presence of hydrogen. The resultant carbon film deposited on a suitable substract has diamond-like structure. The films exhibit the properties of diamond, e.g., hardness, and can therefore be used for abrasion resistant coatings [49]. Graphite composites are materials subjected to isostatic pressing during carbonization, which result in artifacts of high density and no bulk preferred orientation. By control of the conditions, suitable fine-grain optical texture can be obtained, conferring high strength. Carbon electrodes are artifacts that, if the electrodes do not need to be graphitized, use components mainly controlled by cost and availability. For electrodes used in aluminium smelting the filler is calcined, delayed petroleum coke and the binder is coal-tar pitch. The processing is similar to that for graphitic electrodes, with mixing, shaping, prebaking, densifying (in some cases), and heat treatment. The heat treatment temperature, however, is much lower, usually below 2000 K [50].

In summary, in this section, the basic structural features of carbon materials are introduced, and definitions of many carbon forms are given. Two extremes of structural organization are distinguished as from carbonaceous precursors, which pass through a liquid phase on pirolysis (e.g., pitches), being characterized by surface areas often much less than 10 m^2^g^−1^. The later (chars) are formed by those that do not fuse (e.g., wood), being characterized by high microporosity and surface areas above 1000 m^2^g^−1^.

## 3. Catalysis of Carbon Oxidation Reactions

Carbon gasification reactions form the basis of several industrial processes. This section considers a few fundamental aspects of gasification, particularly in terms of catalysis of oxidation reactions, which is assessed by considering mechanisms involving oxygen-transfer stages and topographical changes associated with the catalytic gasification.

The extent to which a particular catalyst will accelerate gasification rate is a complex function of many variables including:The metal concerned,The gasification reaction being studied, and thermal conditions employed,The size of the catalyst particles and their state of dispersion throughout the carbon,The chemical state of the catalyst,The relative amounts of catalyst.

In the majority of earlier investigations, not all of these important parameters were recognized. This is a major reason for the lack of agreement between workers of relative activities of catalysts and activation energies for the catalysed reactions.

A catalyst usually provides an alternative route for the uncatalyzed reaction, with a certain (original) activation energy. The slow stage (rate determining step) of that alternative route has a lower activation energy [51]. Then, this decrease in activation energy, E, results in an increase in reaction rate, which is accompanied by a corresponding change in the pre-exponential factor, A, called compensation effect. E and A obey an equation of the form
m E − ln A = constant(2)
where m is the proportionality constant and A, the pre-exponential factor, is indicative of the density of active sites on the carbon surface. The compensation effect operates as in Figure 4 where there is a crossover through an isokinetic point of the Arrhenius plots of catalyzed and uncatalyzed reactions. An enhanced rate due to both a decrease in activation energy and increase in pre-exponential term is not reported. It is known for the activation energy to remain constant with an increase in the pre-exponential term.

Voluminous literature exists on the mechanisms of the gasification reactions of carbon by oxygen-containing gases such as oxygen, carbon dioxide, water, nitrogen oxide, and sulphur dioxide. Despite the large amount of literature, these reactions are not well understood; this is particularly the case for the C-O_2_ reaction. Studies on the early work were centered around two mechanisms, the oxygen transfer and the electron transfer. In the oxygen transfer mechanism, the catalyst acts as an oxygen carrier that undergoes a redox cycle:MO + CO_2_ → MO·CO_2_(3)
MO·CO_2_ + C → MO + 2CO(4)
where MO represents a metallic oxide. The catalytic compound could include metals, M, as well as oxides, MO. In the electron transfer mechanism, the catalyst ability to accept electrons from the carbon is emphasized, and the interaction between the catalyst and the gas is considered less important.

Metal oxides are generally nonstoichiometric, possessing several point defects such as oxygen ion vacancies and interstitial metal ions and atoms, this deviation from stoichiometry increasing with increasing oxygen partial pressure. Long and Sykes [52] proposed that the metal oxide nonstoichiometry structure could be explained by the electron transfer mechanism. At that time, the ability of metal oxides to catalyze gas reactons, including oxidation of hydrocarbons, was attributed to similar mechanisms. In fact, many of the known metal oxides widely used in catalysis or as heterogeneous catalysts, with a huge variety of composition and electronic and geometric structures, offer a very broad spectrum of properties and behaviors that can result in specific functionalities and chemical activities, allowing them to be used as oxide supports of finely dispersed active metal nanoparticles or directly as catalysts. Usually they have semiconductor character, with band gaps <3 eV, and their capability to exchange oxygen with the liquid- or gas-phase surroundings in a relatively easy way, results in excess electrons that are redistributed on the cation-empty levels, thus changing their oxidation state, which weakens the C-C bond, and allows the CO formation and removal, e.g.,
CO_3_^2−^ + 2C_f_ → 3CO + 2e^−^(5)
2M^+^ + CO_2_ + 2e^−^ → M_2_O + CO(6)
M_2_O + CO_2_ → M_2_CO_3_(7)
where e^−^ is an electron.

For many years, the oxygen-transfer mechanism was the most widely accepted of the two approaches, due to the localised behavior of the catalysts, and because it is also found [53] that the activation energy of catalyzed oxidation of carbon is independent of the concentration of the catalyst present. This is not expected with the electron-transfer mechanism. By the 1990s, it was understood that the active surface oxygen complexes play a key role in the carbon reactions involving oxygen-containing gases. However, many of these surface groups are only stable at temperatures close to room temperature; in other words, for most gasification reactions occurring at higher temperatures, these surface groups do not play a key role anymore. This means that only surface groups with intermediate stability at temperatures higher than 400 °C or 700 °C contribute to the C-O_2_ or C-CO_2_/C-H_2_O reactions, respectively. Highly stable groups may actually be considered as poisons. This type of surface oxygen complex on carbon served to explain major features of the published results on temperature programmed desorption, transient kinetics, and steady-state rate studies of the gas-carbon reactions. A fact that has been forgotten for about 30 years is that the prevailing mechanism for the effects of catalysts on carbon gasification involves carbon bulk diffusion through the catalyst. In order to fully understand catalytic carbon gasification, it is essential to distinguish the initiation phase, in which thermodynamics and the second Fick’s law are essential, and the steady-state gasification process, in which the first Fick’s law geometry and gas surface catalytic reaction are the essential elements to understand kinetics.

All of the gasification reactions of carbon can be catalyzed, mainly by Groups I and II and transition metals. The general understanding of the catalysis process is probably common to all reactions. However, as reaction temperatures and gas pressures differ significantly between the reactions, the energetics and concentrations of the important intermediate adsorbed surface species also differ, thus accounting for different reaction rates, activation energies, and orders of reaction. The efficacy of an inorganic catalyst within a carbon, at least, is a function of the metal, the metal salt (or chemical state within the carbon), the state of distribution and degree of crystallinity within the carbon, concentration, and access to the reacting gas (there may be others). Studies of different catalytic effects using different carbons and different catalysts for different gases at different temperatures with different methods of distribution of the catalyst in the carbon are not likely to give directly comparable results. Hence, the relative efficacies of catalysts or pecking orders differ throughout the literature. A pioneer review is that of Moulijn and Kapteijn [54]. The literature contains several detailed reaction schemes to explain catalysis by oxygen-transfer. Many are postulates because of the difficulty of obtaining analysis of reaction intermediates at reaction temperatures [7,8,23,55,56].

Hence, catalysis by inorganic metals or their oxides enhances the rate of removal of carbon, which forms the carbon oxygen complex. Kinetic studies of different catalysts indicate changes in activation energies and pre-exponential functions. In studies of this complexity, there is no certainty that the mobile surface oxides are all identical, independent of reacting gas, temperature, pressure, catalyst, and catalyst preparation. Hence, differences in kinetic behavior would not be unexpected.

Apart from changing the kinetics of gasification reactions, catalysts are found to accumulate on a carbon surface at imperfections in the graphite basal plane due to the availability of electrons at those imperfections in the lattice [57]. The following modes of surface gasification are attributed to carbon–catalyst interactions.

**Pitting:** As it is shown in Figure 5 [58], the catalyst particle located at vacancies within a basal plane attacks, forming a hexagonal hole that increases in depth due to penetration of the catalyst and can also expand due to the edge recession of the hole.

**Edge Recession:** Catalysts on graphite edge atoms, strongly interacting with C, form a thin catalyst film over the edge carbon atoms, and lead to edge recession (Figure 6) [58].

**Channelling:** It is a mode of attack that occurs when the degree of wetting is less than that observed in the presence of edge recession [57]. During the channel formation, fluidic catalyst particles are left behind on the channel walls. The channel proceeds becoming narrower with a fluted appearance (Figure 7a,b) [58]. Further, it can be straight or can change direction due to structural changes in the carbon gasification (Figure 7c).

Recent thermodynamics studies showed that metal oxide growth is accompanied by strong interfacial bonds with carbons, resulting in edge recession. Noble metals exist as discrete particles at edges, which they attack by the channelling mode [59].

## 4. Nanotechnology

In 1959, at Caltech, R.P. Feynman in his talk <There’s Plenty of Room at the Bottom> pioneered the field of nanotechnology, suggesting a means to develop the ability to manipulate atoms and molecules directly, by developing a set of 1/10-scale machine tools analogous to those found in any machine shop. K.E. Drexler in his 1986 book entitled <Engines of Creation: The Coming Era of Nanotechnology> was the first to use the term nanotechnology. Drexler envisioned the possibility that human-designed molecular robots could replicate themselves just about the same way cells build copies of themselves in order to reproduce. Drexler’s vision of nanotechnology is often called <molecular nanotechnology>. The science of nanotechnology was advanced further when Kroto, Smalley, Curl, and Iijima discovered fullerenes and developed carbon nanotubes. However later, it was B. Fahlman, from NASA, that advocated the most acceptable definition of nanotechnology, stated as the preparation and characterization of functional materials, devices, and systems, as well as the exploitation of novel phenomena and properties at nanoscale, i.e., on the scale of 1 billionth to several tens of billionths of a meter [60]. This definition suggests the presence of a scale issue and a novelty issue: Nanotechnology is concerned with the use of structures by controlling their shape and size at nanometer scale; and nanotechnology must deal with small things in a way that takes advantage of some properties because of the nanoscale.

Old civilizations used nanotechnology to color glass or to make sharp swords [61]. Chemists have worked with micrometer materials for a long time, but it was only by the mid-1980s with the discovery of scanning tunneling microscopy (STM), atomic force microscopy (AFM), and later, high-resolution transmission electron microscopy (HRTEM) that it was possible to identify materials, processes and devices at the nanoscale.

In the last two decades, a wide variety of nanomaterials (sized or structured) were prepared in different forms by either physical or chemical “bottom-up” or “top-down” methods [60,62], which differ in degrees of quality, speed, and cost. The top-down approach is essentially the breaking down of bulk material to get nano-sized particles. This can be achieved by using advanced techniques such as precision engineering and lithography, which have been developed and optimized by industry during recent decades. The bottom-up approach refers to the build-up of nanostructures from the bottom: Atom-by-atom or molecule-by-molecule by physical and chemical methods that are in a nanoscale range (1 nm to 100 nm) using controlled manipulation of self-assembly of atoms and molecules. Self-assembly is a bottom-up approach in which atoms or molecules organize themselves into ordered nanostructures by chemical-physical interactions between them. Positional assembly is the only technique in which single atoms, molecules, or cluster can be positioned freely one-by-one.

The preparation of nanoparticles by bottom-up or top-down techniques leads to great miniaturization of the new materials, and also provides them with new chemical, physical, mechanical, electrical, and optical properties, producing custom-made devices with capabilities not found in bulk materials or in nature, or is even able to replicate some natural processes that have not been currently achieved through synthetic materials. For example, the large increase in the surface area, accompanied by the formation of different crystalline facets, leads to an increase in chemical reactivity especially enhancement of catalyzed chemical reactions. Another example is the exceptional mechanical properties of some carbon nanosctructures like fullerenes, nanotubes, and graphene that are much stronger and lighter than common structural materials like steel. Due to these strength and flexibility, carbon nanotubes, which are memers of the fullerene family, are currently used as composite fibers in polymers and beton to improve the mechanical, thermal, and electrical properties of the bulk product. They also have potential applications as field emitters, energy storage materials, catalysis, and molecular electronic components.

Today, nanotechnology impacts human life every day. The potential benefits are many and diverse. However, because of extensive human exposure to nanoparticles, there is a significant concern about the potential health and environmental risks. These led to the emergence of novel scientific disciplines including nanotoxicology and nanomedicine. Some of the potential benefits of medical nanomaterials include improved drug delivery, antibacterial coatings of medical devices, reduced inflammation, better surgical tissue healing, and detection of circulating cancer cells.

Focusing on the energy domain [63], nanotechnology has the potential to significantly reduce the impact of energy production, storage, and use, thus seriously contributing to a sustainable economic growth. In this context, it seems that the most promising application fields for the energy conversion domain will be mainly focused on solar energy (mostly photovoltaic technology for local supply), hydrogen conversion, and thermoelectric devices. Contributions of nanotechnology to hydrogen production and conversion in fuel cells are described in Section 7.1 and Section 8. Contributions of nanotechnology to sustainable electricity storage via batteries and supercapacitors are provided in Section 10 and Section 11 of this article. We will finish this section with a brief consideration on the contribution of nanotechnology in solar economy. The solar energy source can be used in photovoltaic (PV) technology, which directly converts light into electrical current, solar-thermal collectors, artificial photosynthesis, passive solar technologies, PV solar cells of many types, self-cleaning surfaces, photocatalytic converters, energy efficient windows, electrochromic materials and devices, smart windows, etc. Solar energy materials can be metals, semiconductors, and dielectrics including polymers. They can be bulk-like as well as thin films. Among modern solar energy materials, nanomaterials and nanostructures are of particular interest, namely in electrochromic technologies. Current research is focused on the development of new photoactive materials that can be used to directly convert sunlight (or artificial light) into electricity. PV solar cells consist of semiconductor diodes with two key functions: Separation of electrical charge in energy, and in space. The voltage-current product, or output power, thus depends on light absorption, charge transport, and type of semiconductor junction. This technology based on silicon wafer-based solar cells accounts for more than 80% of the global solar cell market. To improve their performances, new solar cell components include nanomaterials such as carbon nanotubes, activated carbons, fullerenes, fibers, foams, ordered mesoporous carbons, raphene flakes, carbon nanocomposites, quantum dots and wells, etc., which are leading to advanced systems such as the quantum-based and dye-sesitized solar cells. Water splitting by photocatalysts, also known as artificial photosynthesis, is being actively researched, motivated by a demand for cheap hydrogen, which is expected to rise with the new hydrogen economy. The plan of schemes for producing hydrogen through photosynthesis is to alter the normal utilization of reduced ferredoxin, as it occurs during photosynthesis in green plants. Instead of flowing to the enzyme that catalyze carbon dioxide fixation, the electrons from reduced ferredoxin cause reduction of hydrogen ions to hydrogen. This reaction is catalyzed by either of two enzymes that occur in many algae and bacteria: Hydrogenase and nitrogenase. In fact, researchers in several research energy laboratories proved that on concentrating sunlight, high temperature and solar flux are achieved, thus, obtaining hydrogen in a cheap and environmentally friendly way, i.e., able to split methane into hydrogen and carbon. However, the solar photocatalytic hydrogen production is still very limited. That is, again, nanotechnology, by means of its nanoparticles, nanodevices, and precise procedures, is the tool required for the solar hydrogen production in a clean, environmentally friendly, and low-cost way using photocatalytic water splitting.

## 5. Electrocatalysis

In a 14th-century Arabian manuscript, Al Alfani described the <Xerion, aliksir, noble stone, magisterium, that heals the sick, and turns base metals into gold, without in itself undergoing the least change>. Thus, in a chemical reaction, the catalyst enters at one stage and leaves at another. The essence of catalysis is not the entering but the falling out. The word catalysis was coined by Berzelius in 1835. William Ostwald, based on the first law of thermodynamics, was the first to emphasize that the catalyst influences the rate of a chemical reaction but has no effect on the position of equilibrium. It follows that a catalyst must accelerate the forward and reverse reactions in the same proportion.

Catalysis can be designated as homogeneous, if the entire reaction occurs in a single phase, and as heterogeneous if the reaction occurs at phase interfaces. The latter is also called contact or surface catalysis. Most reactions in liquid solutions occur in a unique phase, thus they would not proceed at an appreciable rate if catalysts were rigorously excluded. In general, industrial chemical reactions are run in the presence of solid catalysts. An example of particular relevance is the Pt-catalyzed oxidation of sulphur dioxide to sulphur trioxide, which reacts with water to produce sulphuric acid. A good catalyst should have moderate values for the enthalpies of adsorption of the reactants; moreover, it should possess a great exposed area, often being distributed on the surface of a porous support (or carrier). Its activity may be increased, and its lifetime extended by addition of small amounts (5 to 10%) of substances called promoters. The fluid-phase reactions catalyzed by solids consist of five steps, involving diffusion and chemisorption of reactant species, chemical reaction of adsorbed reactants and fluid-phase molecules, products desorption, and diffusion. In general, one of these steps is much slower than all the others, and only the rate of the slow step needs to be considered.

Carbocatalysis uses heterogeneous carbon materials for the transformation or synthesis of organic or inorganic substrates. One of the most common examples of carbocatalysis is the non-oxidative dehydrogenation of ethylbenzene [64]. In another early example [65], a variety of substituted nitrobenzenes were reduced to the corresponding aniline using hydrazine and graphite as the catalyst.

The discovery of nanostructured carbon allotropes such as carbon nanotubes [66], fullerenes [67], or graphene [68] promoted further developments. These nanomaterials were used to dehydrogenate n-butane [69], to selectively oxidize acrolein [70], to catalytically reduce nitrobenzene [71], and to facilitate the oxidation of alcohols [72].

During the last decade, large progress has been made in the fundamental understanding of the surface chemistry of carbons, opening their use as catalysts for the environmental protection area, selective oxidation processing, hydroprocessing and selective hydrogenation, fine and specialty chemicals synthesizing, etc. [73]. Furthermore, the use of activated carbon (AC) as a support always provides unparalleled flexibility in tailoring their surface area and porosity, as well as their surface functionality properties [73].

A second factor stimulating interest on carbon catalysts is that these materials are one of the first examples for the synthesis of tailored 1D and 2D nanostructures [74]. In this context, the understanding of their surface chemistry and presence of defect sites of carbon nanotubes (CNTs), together with their large availability at low costs allowed a significant progress in the preparation of advanced catalysts. Doping of these materials (particularly with N and B) and tailoring their assembly into three-dimensional has further stimulated their uses in the field of catalysis.

A third pushing factor for the interest on carbon-based catalysts is the worldwide need to develop more sophisticated electrodes for further sustainable utilization of energy.

The recent motivation for the creation of a global-scale sustainable energy system while preserving our environment is one of the most crucial challenges facing humanity today, which, complemented by academic purposes and technical uses in the industry, is pushing catalysis on carbon to electrocatalysis on carbon. An electrocatalytic reaction is an electrochemical reaction with an adsorbed species, which can change the kinetics of the reaction and in some cases also the mechanism. An electrocatalyst (electrode for technological uses) is an electronic-ionic interphase, which accomplishes the surface electrocatalytic reaction, being able to maximally reduce the overpotential required for driving a specific electrocatalytic (electrochemical) reaction.

If a film containing a selective catalyst is attached to an electrode surface, the electrode is said to be chemically modified. Furthermore, the structure, if polymeric, represents a transition from heterogeneous to homogeneous catalysis with the catalytic centers now immobilised on the electrode. If such centers can show redox behavior, electron transport can take place by a hopping mechanism throughout the film. Another type of electrocatalyst is formed when a submonolayer of upd metal is deposited on a substrate. As might be expected, if several monolayers of metal are deposited, the catalytic effects are indistinguishable from the bulk deposit, but a submonolayer shows catalytic properties more typical of a surface alloy. An example is the Pt substrate/Ru submonolayer catalyst, which can be used in the oxidation of both methanol and carbon monoxide to carbon dioxide; this catalyst is bifunctional, with the methanol adsorving on the Pt surface and oxidation being mediated by OH species adsorbed on the Ru. The underpotential deposition metal may also alter the electronic structure of the surface, and this has been proposed as the mechanism whereby a submonolayer of Pb on Pt is effective in promoting the oxidation of formic acid to carbon dioxide.

Charge transfer takes place, for the most part, at the electrode surface and the interaction of substrate with surface strongly influences the overall reaction rate. For pure metals, we can take the flat polycrystalline surface as the reference point, and the reaction rate can then be increased by increasing the effective surface area. In the simplest case, this can be done by roughening the surface, with the current density (referred to the true surface area) remaining constant at a fixed potential, provided mass transport remains sufficiently high. However, in general, this roughening process will also lead to an increase in the number of surface dislocations or defects, both of which can act as active sites for electron transfer. A high density of such sites can also be achieved, for example, by cathodic deposition of the metal, or by preparaton of Raney nickel (from powdered NiAl alloy, from which the Al is removed by dissolution into hot KOH). Metal oxides can also show good electrocatalytic properties, the most familiar case being RuO_2_, which can be deposited on TiO_2_ as a mixed Ti/Ru oxide layer about 1 µm thick. Carbon is also a most useful substrate for electrochemical engineering, with appropriate catalyst coverage to ensure selectivity. The catalytic importance of metal alloys is frequently greater than that of the pure metals, since not only can the electronic properties of one metal be fine-tuned by alloying with another, but bifunctional mechanisms become possible. A simple example is the case in which species A only adsorbs on one component of the alloy and species B on another, the surface of the alloy will then contain neighbouring sites at which A and B separately can adsorb, leading to the possible formation of species such as AB^+^.

Of great technical importance is the influence of formally non-participating solution species on the course of a particular reaction. In a simple but well-known example, we can consider the formation of A^−^ from A and its possible reactions to give AA or AH, the latter by reaction with a proton arising from the solvent water. The reaction can be steered by appropriate choice of supporting electrolyte cation: If a normal cation, such as K^+^ is used, AH is formed, whereas if a large, poorly hydrated cation such as a tetra-alkyl ammonium cation is used, then this will tend to adsorb at the electrolyte surface creating a hydrophobic layer, which prevents protonation and permits the dimerisation reaction to take place.

Along the last two decades, many chemists have shown that the rate of many heterogeneous chemical reactions can be altered by altering the potential between the (metallic) catalyst and a reference electrode. This effect appears to be bound up with changes to both the Fermi level of the catalyst and the electronic work function, both being affected by a change in the electric field at the catalyst surface. The observation of an acceleration of a heterogeneous chemical reaction at an electrode surface by alteration of the electrode potential was, in fact, first reported in 1970 by Vielstich for the decomposition of HCHO at silver in alkaline solution. However, it was only with the investigations of gas reactions at metal catalysts on oxide-ion conducting membranes between 500 °C and 1000 °C, where the potential is maintained between the catalyst on one side of the membrane and a reference Pt/H_2_ electrode on the other, that the generality of the effect became clear. Given that the changes in rate and yield of the catalyzed reaction far exceed the change in electrical current across the membrane, this effect has become known as the <non-faradaic electrochemical modification of catalytic activity>, or NEMCA effect. An example of this effect was observed in the 1990s for the purely chemical oxidation of CO on platinum dispersed on ZrO_2_ as a function of the potential (vs. a Pt/H_2_ reference electrode) on the catalyst material. It could be seen that the basic behavior is similar to the observed for the exchange current density of the hydrogen evolution reaction on different metals of similar electronegativity and structure, as a function of their heats of sublimation. In other words, volcano plots for both systems are similar. A second example can be seen when we plot the rate of chemical decomposition of HCHO to hydrogen and carbon monoxide on gold as a function of the concentration of alkali and the electrode potential on gold by measuring the hydrogen yield through the technique of differential electrochemical mass spectroscopy (DEMS). As yet, molecular mechanisms for the NEMCA effect have not been substantiated; that is, the observed phenomenon may be due to the alteration in the surface coverage of key intermediates as a function of potential. A third example concerns the carbon dioxide electroreduction; this time, the surface chemisty of the carbon surface substantiates the DEMS studies concerning the carbon support degradation, the distribution of products, and the catalytic activity toward the carbon dioxide electroreduction.

There are various applications of electrocatalysis for technological electrochemical reactions, organic electrosynthesis, galvanoplasty, electrode sensors, fuel cells, batteries preparations, and so forth. Figure 8 [75] exemplifies potential pathways for production of important fuels and chemicals, in concert with conventional and green forms of energy production.

The fast and contemporary advances in electrocatalysis are stimulating the conversion of water, carbon dioxide, and nitrogen into the aforementioned products via electrochemical processes coupled to renewable energy. For instance, the currently tested water electrolysis system that works under alkaline conditions not requiring precious metals brings down the cost of water splitting technology, offering a viable way to store energy from solar and wind power in the form of hydrogen fuel, which can be used to produce clean electricity by fuel cells [76,77,78]; thus, this system is an excellent promise for affordable renewable energy. Hydrogen peroxide can also be derived from the oxygen reduction reaction (ORR) as well [79]. Fossil fuel besides its importance as a source of energy and for making chemicals, requires circumvention by synthesizing fuels and chemicals from new feedstocks such as CO_2_ and H_2_O, due to its limited reserves and price. Works on scale-up systems for carbon dioxide electroreduction have been recently reported. This electrochemical approach can convert the gas into chemicals namely carbon monoxide, formic acid, methanol, methane, ethylene, ethane, ethanol, acetic acid, propanol, etc., as well as precursors to polymers and plastics, with the advantage of utilizing excess electrical energy generated from intermittent sources such as solar and wind [80,81]. Likewise, the electrochemical nitrogen reduction reaction (NRR) is currently being intensely investigated as the basis for future mass production of ammonia from renewables. Note that ammonia is the basis of most fertilisers and allows efficient storage and transportation of renewables [82]. The current development of novel electrocatalysts based on controlled surface roughness, atomic topographic profiles, defined catalytic center sites, phase transition along the electrochemical reactions, etc., will push us for a true sustainable energy system.

Figure 9 schematizes various catalyst development strategies [83], which can ideally be addressed simultaneously, leading to the greatest improvements in activity. However, high catalyst loadings can affect other important process such as charge and mass transfer [83], as shown by the plateau in Figure 9, observed at high loadings. Moreover, as illustrated in Figure 9, further strategies can be exploited for catalyst improvement as nanostructuring, adsorption, support modification, polymorphism, confinement, alloying, sheping, doping, intercalation, chemical functionalization, etc. [84,85,86,87].

The effect of particle size on phase transformation and catalytic activity of electrode materials based on nanoparticles is one of the pioneering strategies to develop improved electrocatalysts. Here, we finish this section by briefly elucidating relevant details about the particle size effect on phase transformation and catalytic activities.

### 5.1. Particle Size Effect on Phase Transformations

The interface between two phases in contact has two inherent properties, the double layer capacitance and the faradaic resistance, which we measure experimentally and interpret theoretically. Apart from the faradaic resistance, in the simple case of the metal solution interface, there exists a solution resistance arising from the fact that the potential in solution is always measured far from the interface on the molecular scale, typically at a distance of 10 million nm. The most fundamental equation governing the properties of interfaces is the Gibbs adsorption isotherm [88,89,90] given by
(8)dG = γ dA
where γ is the surface tension, defined in units of force per unit length or energy per unit surface area [N m^−1^ or J m^−2^] and dA represents an incremental increase in surface area, leading to a corresponding increase in the Gibbs energy dG. The surface tension is also the excess surface Gibbs energy per unit area, namely the extra Gibbs energy added to a system as a result of formation of the interface. The notion of an excess surface Gibbs energy is, of course, purely thermodynamic. It is well known that the driving force in chemistry is the decrease in Gibbs energy. Thus, a system wll change spontaneously, in the direction of decreasing surface tension. This leads to two observations: (i) A pure phase always tends to assume a shape that creates the minimum surface area per unit volume. This explains why droplets of a liquid are almost spherical. (ii) When a solution is in contact with another phase, the composition of the interface differs from that of the bulk in such a manner as to minimize the total excess surface Gibbs energy of the system. The second observation represents the essence of the physical meaning of the Gibbs adsorption isotherm. The adsorption of any species in the interface must always cause a decrease in the Gibbs energy of the surface, since it is the reduction in this Gibbs energy that acts as the driving force for adsorption to occur. For liquids or solids involved at the interface, leading to incremental increases or decreases in surface area or to alterations by strain/stress in very small particles, the physical rules are the same, as well as the observed excess surface Gibbs energy [91,92,93].

It is easy to calculate the percentage of atoms on the surface, as a function of the size of the particle assumed as spherical. Knowing its shape, we can establish its volume, which is given by 4 πr^2^Δr, where Δr is the thickness of the shell. The effect of size becomes significant for r ≤ 10 nm, namely where nanoparticles are concerned. It is interesting to note that, small as it may seem, the radius of 10 nm is rather large on the atomic scale. For example, a water droplet of this radius contains about 1.4 × 10^5^ molecules and a similar-sized Pt sphere would contain about 2.8 × 10^5^ atoms.

An isolated nanoparticle in the solution bulk or on the catalyst surface is thermodynamically unstable with respect to a process of merging it with one or more nanoparticles, which could lead to sintering of the nanoparticles into lumps, losing part of their active surface area, and or any inherent catalytic activity they may have had as nanoparticles [94,95].

To prevent or, at least, minimize this issue due to fusing or sintering in systems based on nanoparticles, is to control or restrict the freedom of the nanoparticles to move on the surface of the catalyst. Considering that these particles are free to move in two dimensions, their average thermal velocity can be readily obtained by equating the kinetic energy with the average thermal energy, ν [96,97]
0.5 Mν^2^ = RT(9)

Treating a Pt nanoparticle of r = 10 nm as a large molecule having 2.8 × 10^5^ atoms (cf. Section 5.2), leads to an average thermal velocity of 0.30 ms^−1^, which is really very large on the scale of nanoparticles, for which it was calculated. Thus, it is seen that it will take 33 ns for a particle to move a distance equal to its own radius. This is the upper limit of the average velocity, ignoring all interactions among the particles. This calculation shows that there is more than enough thermal energy (at room temperature) to allow the particles to move and form aggregates, leading to the failure of electrocatalysts based on nanoparticles.

In summary, in analysis of phase transformations in small-volume systems, a size effect should mean the influence of the size of a system on the chemical composition, relative volume, and thermodynamic stability of the phases in the system at constant thermodynamic conditions and a constant composition of the system. Size effects are due to the increase in the fraction of the interface energy in the total energy of the system as its size decreases, which leads to changes in the equilibrium composition and volume of the coexisting phases. These effects can be modeled using methods of equilibrium chemical thermodynamics, where phase equilibrium corresponds to a minimum in the Gibbs energy of the system with allowance for the contribution of the surface energy.

### 5.2. Particle Size Effect on Catalytic Activity

Using density functional calculations, it has been shown that there is a linear relationship between the Gibbs energy of adsorption of a particle on a metal surface and the size of the particle. This relation leads to a volcano-type dependence of the catalytic activity on the particle heat of adsorption [98,99]. However, does a higher energy of adsorption lead to a higher catalytic activity? Based on the related pertinent thermodynamic factors, the existing surface-spectroscopic information, and the volcano plot, it all depends on whether the specific rate constant, here expressed in terms of the exchange current density, which is proportional to the reaction rate at the equilibrium potential, is on the ascending or the descending branch of the reaction rate vs. the bond energy shown in the plot [100,101]. For very small particles, the Gibbs energy of adsorption increases, and the reaction intermediates could be too strongly adsorbed on the surface, reducing its catalytic activity. Indeed, it was found that there is a maximum in catalytic activity for particle sizes in the range 3 ≤ r ≤ 5 nm [102,103,104], which is about where the fraction of atoms on the surface starts rising sharply and the melting points of metals start decreasing very significantly. It should be noted that the ascending branch of the volcano plot is quite convincing, however, in general, on the descending branch there are only oxide-covered metals, with the reaction rate being reduced by several orders of magnitude.

According to Marcus theory, there is a parallelism between the concept of volcano plots in catalysis and outer sphere electron transfer reactions. In summary, volcano plots are a valiant attempt to understand catalytic reactions with the aid of a single descriptor, typically the energy of adsorption of a single intermediate. However, the kinetics of complex reactions are not so simple. Another fact that is important to note is that adsorption in electrochemistry is a replacement reaction that can be described by an equation of the type
RH_soIn_ + n(H_2_O)_ads_ → RH_ads_ + n(H_2_O) _soIn_(10)
where RH stands for an organic species that could be the reactant or one of the intermediates in a reaction sequence. This is referred to as electrosorption. It is characterized by the fact that the Gibbs energy of electrosorption is the difference between that of the energy of adsorption of RH and that of n molecules of water [105,106].
ΔG_ads_ = ΔG_adsRH_ − nΔ G_adsW_(11)

This equation shows that the increase in the Gibbs energy of adsorption results from the decrease in size of the particle but does not necessarily lead to an increase in its catalytic activity, and the effect of size on activity should be tested experimentally for each system [107,108].

Since a nanoparticle is the ultimate case of an ultramicroelectrode, it is appropriate to discuss some of the properties of nanoparticles employing the equations developed for microelectrodes, for calculating the increased rate of diffusion towards an isolated nanoparticle and the corresponding decrease in solution resistance. As discussed by Sequeira et al. [109,110,111], for a single nanoparticle, assumed to be spherical, the limiting current is given by [112,113]:(12)$$j_{L}\;=\;\frac{{\mathrm{nFDC}}_{b}}{\delta }\;=\;\frac{{\mathrm{nFDC}}_{b}}{r}$$
where the radius of the particle plays the role of the Nernst diffusion-layer thickness, δ in the case of semi-infinite linear diffusion. For a nanoparticle of r = 5 nm, taking n = 1; D = 6 × 10^−6^ cm^2^s^−1^; and C_b_ = 1 × 10^−6^ molcm^−3^ yields, according to Equation (11) it results a very large limiting current density of j_L_ = 1.16 A cm^−2^, for a rather dilute solution (1.0 mM) of the reactant [114].

It may be added here that stirring will have no influence on the limiting current density calculated above, because the Nernst-diffusion layer thickness is δ ≥ 5 µm. This can be seen from the equation for the limiting current density, considering both stirring and radius of the nanoparticle, which is given by
(13)$$j_{L}\;=\;{\mathrm{nFDC}}_{b}\left(\frac{1}{\delta }+\;\frac{1}{r}\right)$$

The solution resistance for the same nanoparticle is given by
(14)$$R_{s}\;=\;\frac{r}{k}=\;\frac{5\times 10^{-7}\;\mathrm{cm}}{0.01\;{S\;\mathrm{cm}}^{-1}}=5\times 10^{-5}\;\Omega {\mathrm{cm}}^{2}$$
where a moderate specific conductivity of k = 0.01 S cm^−1^ has been assumed. At a current density of j_L_ = 0.1 A cm^−2^, the resulting potential drop is then given by
j R_s_ = 0.1 A cm^−2^ × 5 × 10^−5^ Ωcm^2^(15)

This looks like an ideal situation for conducting electrochemical measurements at current densities far below the limiting current density, but there is a problem. The surface area of a nanosphere of 5 nm radius is about 3 × 10^−12^ cm^2^. Hence, at 0.1 A cm^−2^, the total current is only 3 × 10^−13^ A. This is measurable but not useful for any device.

In the time-honoured method of preparing reversible hydrogen electrodes employing platinized-platinum, the real surface area of the electrode is increased by roughening it, without really increasing the physical dimension of the electrode. In this way, the effective exchange-current density can be increased by as much as two or three orders of magnitude, without changing the true catalytic activity of the surface. This approach is routinely applied successfully for commercial batteries and fuel cells. Improved performance is achieved using very high surface area materials.

Methods of producing such electrodes with carbon as a substrate have been developed over the years, and the recently produced ones attain specific areas higher than 2.5 × 10^7^ cm^2^ g^−1^. Such materials could be mixed with Pt or Pt-Rh alloys, preparing similarly high surface area catalysts [115,116]. There are also several methods of measuring the surface area of porous materials, but there seems to be some uncertainty regarding the true area where the electrochemical process can take place [117].

The determination of the catalytic activity of single nanoparticles, as a function of size, is a very delicate matter since the area of such particles is of the order of 10^−12^ cm^2^, and any error in the measurement (or better, the estimation) of the surface area can lead to a major error of 0.1 A cm^−2^ or more. In addition, the volume-to-surface area ratio in such measurements is extremely high, i.e., a meaningful determination of the electrode inherent catalytic activity, can be an insurmountable challenge. This leaves the question of the dependence of catalytic activity on the size of the nanoparticle open to debate. On the other hand, it is a fact that by using nanoparticles we can prepare effective high-surface area electrodes, thereby increasing the catalytic activity. Whether this is due to an increase in the intrinsic catalytic activity associated with the small size, or just to the increase in electroactive surface area of the electrode, has no interest from the practical point of view, for example in the design of the better carbon anodes in fuel cells.

Readers interested in physics of nanoparticles in general can consult books cited in references [117,118,119,120,121,122,123].

## 6. Aluminium Smelter Technology

Aluminium is the most abundant metal in the Earth’s crust, and the third most abundant element, after oxygen and silicon. It makes up about 8% by weight of the Earth’s solid surface. Rapid dissolution of aluminium in any acid and alkaline solutions is always found, but around neutral pH, the oxide formed is very dense and non-conducting, and oxidation is effectively stopped after a thin layer of about 5 nm has been formed. This thin layer of oxide permits aluminium to be used as a construction material and in many other day-to-day applications. There are, of course, additional ways (e.g., anodizing and painting), to protect aluminium that is exposed to harsh environments, beyond the protection afforded by the spontaneously formed oxide film. However, the unique feature of this metal is that it repassivates spontaneously when the protective layer is removed mechanically or otherwise, as long as the pH of the medium in contact with it is in the appropriate pH range (4.0 < pH < 8.6). Aluminium is both light and strong, may readily and cheaply be treated by anodizing to retard corrosion, has a strong affinity for oxygen, and is the principal alternative to copper as a conductor of electricity. Moreover, the known reserves of aluminium ores are relatively high. Aluminium metal is too reactive chemically to occur in nature as a metal. Instead, it is found combined in over 270 different minerals. Due to its reactivity, aluminium metal is a modern metal with an annual production currently approaching 40 million tons. Up until the late 19th century aluminium metal was considered a pressure metal and most of the metal at that time was produced by metallothermic reduction (K or Na) of anhydrous aluminium chloride. Bauxite, which contains 20–30% aluminium as hydrated oxide and hydroxide, is the principal ore. It is chemically purified by the Bayer process to 98+% pure aluminium oxide (alumina). In 1887, Karl Bayer discovered that aluminium hydroxide that precipitated from alkaline solution is crystalline and can be easily filtered and washed. By dissolving aluminium from bauxite, the rest can be separated from the liquor as solids. The aluminium hydroxide is then dried and calcined to give high-quality alumina. These inventions sealed the fate of aluminium, and by 1890, the cost of aluminium had tumbled about 80%. The process is still in use today all over the world.

Since aluminium is quite an active metal, the traditional smelting technique used for iron did not work, and electrolysis, with the significant development of the electrical generator, was the only practical method to enable the electrolytic aluminium production. Reduction of Al (III) from aqueous solution was also impossible since hydrogen would be evolved first even from strongly basic solutions. The solution to these restrictions was discovered in 1886 independently by Hall in the United States, and Héroult in France. Charles M. Hall (1863–1914) founded the Aluminium Corporation of America (Alcoa). Paul L.T. Héroult (1863–1914), with Henri Le Châtelier, worked on the aluminium smelting problem, and developed furnace techniques for production of steel. The Hall–Héroult process makes use of the solubility of alumina (Al_2_O_3_) dissolved in a molten cryolite-based electrolyte (Na_3_AlF_6_), to give a conducting solution from which molten aluminium (m.p. 660 °C) can be obtained at a steel-reinforced carbon at the bottom of the electrolytic cell; simultaneously, oxygen from the alumina would react with the carbon anode, forming carbon dioxide.

The first manufactured carbon electrodes were in the shape of rods initiated for the need for carbon used for electric arc lights. By the beginning of the last century, the aluminium industry demanded larger electrodes as the cell size and amperage started to increase. Rectangular anodes of 25 × 25 cm cross-section were operating at over 6 A cm^−2^, and at the same time it became profitable for the aluminium companies to produce anodes themselves and reuse the 20–30% butts from the spend anodes as raw material for new. A fundamental change came when the Soderberg anode was developed in Norway. In this concept, briquettes made of coke and pitch were introduced at the top, and as the anode was gradually consumed and lowered into the electrolyte, the briquettes became soft and were eventually baked into anodes by the heat generated in the cell before being consumed when it reached the electrolyte 1,2 months later. The Soderberg anode started in 1924 and became widely used in the world in the 1940s and 1950s. In the early 1950s, a continuous prebake anode was developed by VAW. In this process, new prebaked anodes were glued onto the top of the spent anodes and electrical connection points, or studs, were changed from the spent anode to the new during operation, forming one large anode per cell. Prebake anode technology is the dominating technology in the world today. The latest development in anode technology is normally to make one to three slots in the anode to drain away the anode gas being produced. The slots efficiently drain away the gas, enabling high current efficiency and low cell noise in modern high-amperage cells. The slots are normally made by cutting or are already introduced in the green state in the anode mold.

Concerning the cell construction, there are mainly two types of cells, a Soderberg and a prebake cell. The main difference between these two types of cells is the anode. The cells are then subdivided based on where the alumina feeding takes place (side or center). The Soderberg cells are also named based on how the studs (anode collector bars) are oriented in the anode (sideways or vertically). Generally, a modern prebake cell is much larger than a Soderberg cell; therefore, the width/length ratio of the cathode might be a little bigger for the Soderberg cell. In the modern times, the need for improving the cathode life, and for higher cathodic current densities and lower energy consumption, required changes in the carbon quality from anthracitic to semi-graphitic/graphitized until fully graphitized today. The side lining of today may contain graphite or preferably silicon carbide to obtain high heat loss in high-amperage cells, but a frozen ledge is still needed for protection.

Usually, the most common electrolyte contains alumina with some calcium fluoride, and an excess of aluminium fluoride, AlF_3_, over the molten cryolite composition. The mole ratio NaF/AlF_3_ is defined as the cryolite ratio while the mass ratio of NaF/AlF_3_ is called the bath ratio. Numerically, the bath ratio is half the cryolite ratio. Pure cryolite melts at 1012 °C but alumina and aluminium fluoride lower the melting point. Owing to a small amount of calcium oxide impurity in the alumina, calcium fluoride attains a steady-state concentration of 3–8% at which level calcium is codeposited in the aluminium and emitted in the off-gases at a rate equal to its introduction. Calcium fluoride lowers the liquidus temperature about 2.9 °C per weight percent calcium fluoride. Sometimes, lithium fluoride is added to increase the electrolyte conductivity and the current efficiency, and further lower the liquidus temperature. Today, the electrolyte contais mainly cryolite typically with the addition of 10–11 wt.% AlF_3_ to lower the melting point, 2–4% alumina, and about 5% of CaF_2_ (impurity from the alumina). Some smelters also add small additions of LiF, KF, and MgF_2_ to the electrolyte. Then, an electrolysis cell can be operated at about 950–960 °C.

Molten cryolite ionizes
Na_3_AlF_6_ → 3Na^+ −^ + AlF_6_^3−^(16)

The hexafluoroaluminate ion undergoes further dissociation with typically the addition of
AlF_6_^3−^ ⇔ AlF_4_^−^ + 2F^−^(17)
(18)$$K\;=\;(a_{{\mathrm{AlF}}_{4}^{-}})\;({a^{2}}_{F^{-}})/a_{{\mathrm{AlF}}_{6}^{3-}}\;=\;4\alpha^{3}/[(1-\alpha )(1\;+\;2\alpha {)}^{2}]$$

The mechanism and the degree of dissociation, α (about 0.3 at the cryolite composition), of the hexafluoroaluminate ions have been developed from cryoscopy, density, viscosity, and Raman spectroscopy [124,125,126].

Since rapid dissolution and high activity of alumina are desirable, additives are usually limited to less than 10 weight percent. When alumina dissolves, a strong chemical interaction takes place between the solute and the solvent causing melt properties to change [127]. Complex, rather bulky oxyfluoroaluminate ions, Al_2_O_x_F_Y_^6−2x−y^, are likely rather than the simple aluminates such as AlO_2_^−^, AlO^+^, Al_2_O_4_^2−^, or AlO_3_^3−^. Gilbert et al. [6] found by Raman spectroscopy that all single oxygen atoms had bridging Al-O-Al bonds. Reactions (19) and (20) present dissolution mechanisms consistent with the above.
4AlF_6_^3−^ + Al_2_O_3_ → 3 Al_2_OF_6_^2−^ + 6F^−^(19)
2AlF_6_^3−^ + 2 Al_2_O_3_ → 3 Al_2_OF_6_^2−^(20)

The solution mechanism of eqn. (19) would be favored by low alumina (<1%) concentration and low bath ratio (<1.2) while eqn. (20) would be favored at higher alumina concentrations and higher bath ratios.

Although Na^+^ ion is being the main current carrier in the electrolysis cell, aluminium is the thermodynamically preferred product, being favored over sodium in the industrial electrolyte compositions. As there is no evidence that Al^3+^ ions are present, all of the aluminium in the melt is bound in different anionic complexes. Al-O-F takes part in the anode reactions, so the most probable cathode reactions involve the remaining aluminium-containing ions AlF_6_^3−^ and AlF_4−_.
AlF_6_^3−^ + 3e^−^ = Al+6F^−^(21)
AlF_4_^−^ + 3e^−^ = Al+4F^−^(22)

Reaction (21) may be less favored because of the stronger electrostatic repulsion of AlF_6_^3−^ ions from the cathode. In any case, these reactions explain why the electrolyte close to the cathode contains a high concentration of F^−^ ions.

In the electrolytic cell, the direct current goes from the carbon anode to the carbon cathode, facilitating alumina dissolution and aluminium production that facilitates alumina dissolution. Since the temperature of anodes differs by height, they must have good mechanical properties to avoid breakage and separation of individual parts of anode into the bath. Components of high quality for the production of anodes are petroleum coke (60–70 wt.%), coal tar pitch (14–17 wt.%), and anode scrap (15–20 wt.%).

The calcined petroleum coke, obtained by carbonization of heavy oil fractions and oil residue, is used as filler in the production of carbon anodes. It must be of optimum density, good electrical conductivity, and appropriate strength to ensure anode thermal characteristics and stability during the electrolysis process [128,129].

Coal tar pitch is mainly used as a binder in the production of electrodes. It binds the coke particles by entering the pores and filling the cavities between them. Viscosity, penetration ability, and chemical reactivity define its good binding properties [130,131].

Green anode scrap is created after rejection of the first anode mass until the appropriate mass temperature is reached. In the electrolysis process, when replacing anodes from time to time, the appearance of unused portions of the anodes as baked anode scrap is unavoidable. It can also be created after damage of baked anodes in transportation or from rejected waste [4].

The primary anode reaction may be represented simplistically by
C + 2O^2−^ − 4e^−^ → CO_2_(23)
but oxygen is easily complexed in the electrolyte, thus the anode reaction involves such complex ions as
Al_2_O_2_F_4_^2−^ + 2AlF_6_^3−^ + C → 4AlF_4_^−^ + CO_2_ + 4e^−^(24)
2Al_2_OF_6_^2+^ + 2AlF_6_^3−^ + C → 6AlF_4_^−^ + CO_2_ + 4e^−^(25)

Most of these initial oxygen-containing ions are transported through the double layer and discharged with comparatively little overpotential [132]. This oxygen is chemisorbed on the carbon surface as proposed by Blyholder and Eyring [133] for ordinary combustion, forming C_2_O (Figure 10a).

This breaks down slowly to CO gas. Accumulating C_2_O removes sites for further oxygen discharge. As more sites become occupied, it takes more energy (overpotential) to deposit oxygen. Finally, C_3_O_2_ forms, (Figure 10b) which desorbs rapidly as CO_2_ forming fresh carbon sites for oxygen deposition.

Considering the cathode and anode primary reactions, which can be simplified as:4Al^3+^ + 12e^−^ → 4Al(26)
6O^2−^ + 3C → 3CO_2_ + 12e^−^(27)
the overall cell reaction can be written as:2Al_2_O_3_ (dissolved) +3C (s) → 4Al(l) + 3CO_2_ (g)(28)

A typical aluminium reduction cell is shown in Figure 11.

The total cell potential in the Hall-Héroult process is comprised of three different contributions, which are: (i) The reversible cell potential for the overall reaction (influenced by cell temperature and alumina concentration); (ii) the electrode overpotentials required for the occurrence of the anode and the catode partial reactions at a reasonable rate (depends on operating conditions); and (iii) the ohmic drop (due to the resistance in the electrolyte and in the electrodes). The reversible cell potential is about −1.215 V; for an ideal cell process, and since 3 Faradays of charge are required per mole of the aluminium, we would expect an electric power consumption of about 350 kJ/mol. The ohmic drop in the electrolyte is about −1.535 V, a large value that may be due to the large interelectrode gap, and the undissolved alumina. The ohmic drop in the anode is about −0.420 V, and in the cathode is about −0.680 V, values that may be ascribed to the low carbon conductivity, gas bubbles on the anode, and impurities such as P and V, which show variable oxidation states, being reduced at the cathode and then reoxidized at the anode, consuming current without production any metal. The anode and cathode overpotentials are about −0.510 and −0.080 V, respectively. Thus, the total working cell potential is about −4.450 V. Added to this is the energy cost of heating the cell to 1000 °C, purifying and drying the bauxite, and preparing the carbon anodes. A typical cell size was about 10–20 kA in 1914, and 50 kA in 1940. Today, new cells operating at 400 kA are quite normal, and some companies are now working with concepts to reach above 600 kA. A typical cell house will contain about 400 cells arranged in series on two lines, with each 6 × 8 m cell having a total anode area of 30 square meter. The optimum current density is around 1 A cm^−2^, giving a total cell current of 300 kA, and this requires a cell potential in the range from −4.0 to −4.5 V. The cell potential, of course, depends on alumina concentration since this determines the concentration of electroactive species at both electrodes. It drops to just below −4.0 V after addition of alumina to 6% and rises to about −4.5 V before the onset of an anode effect. Hence, the cells are arranged in the cell house to produce the minimum magnetic field. The primary anode reaction always leads to some loss in current efficiency, and in most cells, the aluminium current efficiency is only 85–90%. From these data, the energy requirement may be estimated to be over 14,000–16,000 kWh per ton of aluminium, and we can also calculate that the cell house described would produce 140,000 ton/year.

Modern studies, including fluid dynamics of the Al cell and its significant factors in cell life and current efficiency, combined heat flow and electric current flow extended to ledging and crusting, and other studies are improving the performance of aluminium production. The Light Metals series of books published by The Metallurgical Society of AIME, and others, also provide a wealth of both fundamental and practical information on these topics. The final remark has to be that the Hall–Héroult process has shown to be a very robust and profitable process, surviving any attempt to be replaced.

## 7. Electrochemical Kinetics

More than ever before, electrochemists now address problems of general scientific interest and use a large variety of other techniques, while researchers in many fields routinely resort to electrochemical measurements to obtain essential information. Consequently, electrochemistry has become an important facet of modern science, especially in surface and materials science. In this context, the basic physical principles that govern electrochemical systems, including bulk electrochemistry and interfacial electrochemistry, have been intensively researched during the last two decades. This is particularly obvious in the case of electrocatalysis with the aim of increasing the performance of novel electrocatalysts for a required carbon electrode process. Electrochemical, chemical, thermal, hydrothermal, solvothermal, annealing, doping and codoping, pirolysis, structure modification, coordination with other heteroatoms, introduction of defects, etc., are being largely reported to develop this field. An appropriate understanding of the resulting modifications requires investigations by electrode kinetics using fast and slow electrochemical techniques as reported in the open literature [134,135,136,137,138,139,140,141,142,143,144,145,146,147,148,149,150,151,152,153]. By using different types of carbons (glassy carbon, carbon felt, carbon paper, graphite, graphite-reinforced carbon, carbon fibers, reticulated vitreous carbon, carbon nanowires, carbon nanotubes, carbon dots, graphene flakes, graphene/graphene oxide foams, carbon decorated with nanoparticles, doped with heteroatoms, etc.), prepared or modified as several reported above, it has been possible to obtain significant information about current relevant electrode processes [154,155,156,157,158,159,160,161,162,163,164,165,166,167,168]. In this context, here we will discuss some emerging heterogeneous electrocatalysts and relevant electrochemical transformations involving water, hydrogen, oxygen, hydrogen peroxide, carbon dioxide, and nitrogen, whose address would allow the sustainable production of many fuels and chemicals.

### 7.1. Hydrogen Reactions

The rapid depletion of our fossil fuel reserves and our ever-increasing energy demand have led us to a vigorous search for non-fossil energy sources (nuclear, solar, wind, tidal, geothermal, and ocean thermal energy). However, these sources are handicapped by their not being portable, and by not offering the logistic convenience and facility of oil. Therefore, a new energy carrier, the hydrogen secondary energy carrier, has been uunder serious consideration around the world, particularly in this century. Due to technological problems, the concept of nuclear power to hydrogen has not been vigorously pursued, and conversion of electricity into hydrogen has been more relevant. In fact, the electrolytic hydrogen technology has been used for industrial production of hydrogen for more than 100 years, and one of the most studied problems in electrode kinetics is the reduction of hydrogen ions to form hydrogen gas [169,170,171,172,173,174]. One of the most striking features of the experimental results is the enormous range in the exchange current density with electrode material. Hydrogen evolution at a platinum electode has an extremely high exchange current density when compared with its value at a lead electrode. Furthermore, the apparent cathode transfer coefficients vary from less than 0.5 to about 2.0. These features can be understood, at least qualitatively, in terms of a simple model, as described below. An advanced catalyst for electrochemical hydrogen evolution reaction (HER; 2H^+^ + 2e^−^ → H_2_) should reduce the overpotential, and consequently increase the efficiency of this important electrochemical process [174]. The HER is a classic example of a 2-electron transfer reaction with 1 catalytic intermediate, H*, where * denotes a site on the electrode surface. In electrode kinetics, its reaction rate vs. overpotential relationship can be expressed by the Butler-Volmer equation [76,77], which can be simplified to the well-known empirical Tafel relation for overpotentials higher than 50 mV. Then, two important parameters (the Tafel coefficient, b, and the exchange current density, j_0_) can be obtained in which the dominant reaction mechanism of HER processes can be revealed based on the value of b. In general, for the HER under acidic conditions, at room temperature and 1.0 atm., H^+^ in the electrolyte obtains an electron and subsequently forms an adsorbed hydrogen atom on the active site of the electrocatalyst; this is referred to as the Volmer reaction (Volmer step 29). As for the subsequent reaction step involving the desorption process, this can proceed through two different mechanisms to produce hydrogen. In the case, in which the adsorbed hydrogen atom coverage is relatively low, the adsorbed species prefers to react with H^+^ and electrons to evolve H_2_ through an electrochemical desorption process that is referred to as the Heyrovsky reaction (Heyrovsky step 30). As the H* coverage is relatively high during the process of reaction, the dominant desorption processes will be transferred from above electrochemical desorption to recombination process between adjacent H^*^, which is also referred to as a Tafel reaction or a chemical desorption reaction (Tafel step 31). If b is −30 mV/dec, it means that step 29 is fast, and the chemical desorption process is the rate-determining step (Tafel step 31). If b is −40 mV/dec, this also indicates that step 29 is fast, but that the hydrogen desorption rate is relatively slow and therefore hydrogen is evolved through the electrochemical desorption process as the rate-determining step (Heyrovsky step 30). Lastly, in cases in which b is −120 mV/dec, this indicates that the first step is slow (Volmer step 29) regardless of whether hydrogen is evolved through the electrochemical desorption reaction or the chemical desorption reaction. As for the value of j_0_, this is another important parameter to evaluate the intrinsic activity of electrocatalysts in which the electrochemical reaction rate at reversible conditions can be obtained. In general, small b and large j_0_ values are desirable for ideal electrocatalysts. In summary, the HER may occur through either the Volmer-Heyrovsky or the Volmer-Tafel mechanism [83,175,176].
Volmer step: H^+^ + e^−^ → H*       (b = −120 mV)(29)
Heyrovsky step: H* + H^+^ + e^−^ → H_2_ + *  (b = −40 mV)(30)
Tafel step: 2H* → H_2_ + 2*        (b = −30 mV)(31)

Tafel slope, b, is the inherent property of a catalyst, and is determined by the rate limiting step of HER; the b values quoted are for 25 °C and a symmetry coefficient of 0.5 [175]. As for alkaline conditions, the adsorbed hydrogen atom is generated from the electrochemical reduction of H_2_O, which is more sluggish than the reduction of H^+^ in acidic conditions because the H-O-H bonds need to be broken before adsorbing hydrogen. Therefore, it is more facile to evolve hydrogen in acidic solutions than in alkaline soluions. However, because most electrocatalysts are unstable in acidic solutions due to poor corrosion resistances, strategies such as metal hybrids or carbon-coated structures need to be developed to enhance the stability of HER electrocatalysts.

The hydrogen adsorption free energy, ΔG_H_ [177,178], plays a key role in electrode kinetics. If hydrogen binds to the surface too weakly, step 29 will limit the overall reaction rate. In the absence of double-layer effects, the anodic and cathodic transfer coefficients should be near 0.5. Kinetic parameters for hydrogen evolution on Pt, Hg, Zn, Sn, Cd, Cu, Fe, and Ni in contact with 1.0 M HCl at 20 °C are consistent with step (29) being rate limiting. If hydrogen binding is too strong, the desorption (Heyrovsky/Tafel) step will limit the rate. Thus, if step (30) is rate limiting, we expect that the cathodic transfer coefficient should be approximately 1.5 and the anodic transfer coefficient should be about 0.5. Step (30) is probably rate limiting for Pt, Au, Mo, and W in contact with 1.0 M HCl at 20 °C. This leaves the apparent transfer coefficient of 2 for Pd as an anomaly. Palladium behaves anomalously with hydrogen in other kinds of experiments. The mobility of hydrogen atoms in bulk palladium is known to be anomalously high so it may be that the reaction goes via step (31), which is rate limiting. In this case, if k is the forward rate constant for step (31) and C is the surface concentration of adsorbed hydrogen atoms, the net current for a cathodic overpotential should be given by j = 2FAkC^2^. For small overpotential and small surface concentration of H atoms, the electron-transfer step is nearly at equilibrium, so that the rates of the cathodic and anodic steps are nearly equal. Thus, the surface concentration is C = [H^+^] exp F ñ/RT and the net current is given by j = j_0_ exp (−2Fñ/RT), where ñ is the overpotential. Thus, the apparent cathodic transfer coefficient for this mechanism is expected to be about 2, consistent with the results for Pd in contact with 1.0 M HCl at 20 °C. In summary, a necessary but insufficient condition for an active HER catalyst is ΔG_H_
≈ 0 [177,178]. In plotting experimentally measured exchange current densities or overpotentials for a wide range of catalyst materials against ΔG_H_ at the appropriate coverage calculated from density functional theory (DFT), volcano relationships emerge (see Figure 12C), illustrating the Sabatier principle [179,180,181,182,183,184,185]. Understanding how to control binding energies of reactive intermediates on a surface is the key to designing materials with improved performance. In this context, it is convenient to understand, at least qualitatively, the correlations between exchange current densities or overpotentials and bonding energies, i.e., the volcano-type plot. Accordingly, let us note that the role of a heterogeneous catalyst is to adsorb the reactant or intermediate and transform it to a species that can undergo the desired chemical reaction more readily. If the bonding enegy of adsorption is very low, the extent of adsorption will be very small. As the bonding energy increases, the fractionl surface coverage also increases, and the adsorbed species can be modified and activated by its bond to the surface. However, this can be overdone. Beyond a certain point, the coverage approaches saturation, and the rate of reaction can no longer increase with increasing energy of adsorption. Also, if the bonding energy to the surface is too high, the adsorbed intermediate or its product may stick to the surface, effectively poisoning it. In short, the energy of adsorption must be high enough to attach the reactant to the surface in sufficient amount, yet low enough to allow it to react there and release the product to the solution. This shows that the best catalyst is one giving rise to an intermediate value of the bonding energy for adsorption. However, caution is appropriate when making such correlations. Often some of the properties are correlated (e.g., the work function, the potential of zero charge, and the heat of adsorption are nearly linearly related), but finding a numerical correlation between any two quantities does not necessarily prove that one is caused by, or is directly related to, the other. On the other hand, support for the volcano plot may be questioned for the data given on the right-hand side of the plot, where the exchange current density or overpotential declines with increasing M-H bond strength, and this can be understood because the metals involved all have an oxide layer that is very hard to remove, so that the rate of hydrogen evolution may have been measured on the oxide rather than on the bare metal, in which case, the M-H bond strength may not be relevant.

The most effective HER electrocatalysts ar Pt group metals [186,187,188], but for the last 2–3 decades attention is being particularly oriented to materials that are abundant and of lower cost. Nanoparticles with abundant catalytic edge sites, excellent chemical, physical, and mechanical properties, coupled to carbon materials, namely graphene and other networks, are allowing the development of electrodes with good electrocatalytic activity and durability for the HER [189,190,191,192,193,194,195].

An excellent example of these new electrocatalysts are MoS_2_ –based HER catalysts [83], that for many years were considered as inactive for the HER [196]. However, DFT calculations revealed the high activity of exposed MoS_2_ edge sites to hydrogenases and nitrogenases in nature [195,197]. These motivated the development of several MoS_2_ catalysts with many exposed edge sites [198,199,200,201,202]. Other approaches were explored [194,195], namely by preparing MoS_2_ nanoparticles on reduced graphene oxide (RGO) nanosheats [203,204].

Based on theory, an optimal hydrogen oxidation reaction (HOR) catalyst should also exhibit a ΔG_H_
≈ 0 and reside at the top of the same volcano plots. As such, we would expect Pt to be the best pure metal catalyst for both the HER and HOR in acid, which is in fact observed experimentally [205,206]. However, MoS_2_ catalysts show much poorer HOR activity compared to the HER [198]. To understand these differences [206,207] for the HOR compared to the HER, it is necessary to consider coverage effects, metallic surface states, surface structures, catalyst stoichiometries, etc. For example, the ΔG_H_ of metallic Pt has little dependence on hydrogen coverage, whereas that of MoS_2_ shows significant variation [197]. Also, the metallic surface expected during HER conditions is quite different from the metal oxide/hydroxide surface that can form during HOR conditions [207]. As a result, we all continue searching for Earth-abundant materials with electronic conductivity and nearly constant ΔG_H_ that may lead to advanced electrocatalysts for both the HER and the HOR.

### 7.2. Oxygen Reactions

Oxygen evolution reaction (OER) is a crucial reaction for many energy technologies such as high efficiency water electrolyzers, or photo-driven water splitting, regenerative fuel cells, and advanced rechargeable metal-air batteries. Accordingly, high performance catalysts are urgently needed to speed up the OER, lower the high overpotential required to drive the reaction and reduce the energy consumption.

The OER is a more complex reaction than the HER, involving the transfer of four electrons. Let us assume a mechanism (one of many possible) for this reaction. The first step would be charge transfer as a rate-determining step involving one electron and a very low coverage by adsorbed OH. This could be followed by a further charge-transfer step involving another electron and the combination of the OH^−^ with the adsorbed OH, leading to an increased partial coverage by adsorbed O. Then, this can be followed by atom–atom recombination, yielding O_2_. Considering this atom-atom recombination as rate-limiting step, the two preceding steps are at quasi-equilibrium, being easy to show that for this mechanism, the transfer coefficient is 4, the Tafel slope is 15 mV/dec, and the reaction order at constant potential is 4. It is important to note that, in a complex reaction sequence, there can be more than one type of adsorbed intermediate on the surface, and some steps may involve the transformation of one kind of adsorbed species to another, by either a chemical or an electrochemical route.

Two aspects of the OER are common to all electrodes studied so far: (i) The exchange-current density is low, of the order of 10^−10^ A/cm^2^ or less, and (ii) a reversible oxygen electrode operating at or near room temperature has not yet been found. At sufficiently high temperatures (say, in molten salts, at about 600 °C, or with high temperature solid electrolytes operating at around 1000 °C) the kinetics of the reaction can be sufficiently accelerated to make reversible oxygen electrodes operate.

An important point that is often ignored is that by the time the reversible potential for oxygen evolution is reached, an oxide layer has been formed on all metals. At more anodic potentials, where measurements can actually be conducted to follow the oxygen evolution, the oxide film may be several molecular layers thick. In other words, the OER never occurs on the bare metal surface.

Oxygen evolution can occur readily on the electronically conducting oxides such as ruthenium dioxide and the oxides formed on platinum and iridium. On semiconducting oxides such as NiO, and the oxides formed on tungsten and molybdenum, the OER can still occur, but it may be associated with pitting on the one hand and further build up of the oxide layer on the other, causing poor reproducibility and making the interpretation rather dubious. On insulators, such as alumina and the oxides formed on the valve metals (Ti, Ta, and Nb), oxide formation is the main reaction occurring during anodic polarization and the current either decays to zero with time or reaches a constant value, at which the rate of dissolution of the oxide is equal to the rate of its electrochemical formation.

On noble metals (e.g., Pt, Ir, Au), the region of potential in which the OER is studied is far removed from the potential of zero charge in the positive direction. Thus, only negatively charged impurities will be heavily adsorbed, and the nature of the anion in the electrolyte will influence the measurements substantially. Also, most organic molecules are rapidly oxidized at these potentials, so that the requirements for solution purification are much less severe than in the case of hydrogen evolution.

For many years, iridium dioxide and ruthenium dioxide have been considered to be the state-of-art electrocatalysts for the OER. However, from the view of the shortage and cost, these noble metal oxides require urgent substitution by cost-effective catalysts based on Earth-abundant and low-cost transition metals (bifunctional oxygen electrocatalysts, non-stoichiometric oxides, nanostructured catalysts, NiCo layered double hydroxide nanosheets, graphene-supported MnO_2_ nanowires, CuCo_2_O_4_ nanoparticles on nitrogenated graphene, etc.). Among several strategies designed to increase the electrocatalytic activity of the OER, the three most common include the decrease of the catalyst particle size, the formation of a porous structure, and the couple of carbon-based materials. Therefore, some novel nanomaterials, such as nanoparticulate functional materials, 3D hybrids of N-doped porous carbon nanosheet/CoNi alloy-encapsulated carbon nanotubes, ordered mesoporous nickel sphere arrays, porous trimetallic foams, monodispersing NiFe alloyed nanoparticles in 3D strongly linked sandwiched graphitized carbon sheets, etc., have recently been synthesized and used to enhance the electrocatalytic activity [208,209,210,211,212,213,214,215]. Among these novel nanomaterials, the 3D porous carbonaceous electrodes can be used as efficient electrocatalysts for OER. The application of N, O-dual doped graphene-carbon nanotube (NG-CNT) hydrogel film as a catalyst for OER revealed that NG-CNT hydrogel film electrode can deliver a 5 mA cm^−2^ current density at an overpotential of 368 mV, demonstrating strong stability in alkaline solutions [216].

On the other hand, a 3D N-doped carbon film assembled from graphene and graphitic carbon nitrate ultrathin nanosheets on the frameworks of carbon fiber papers (G-C_3_N_4_) delivered a current of 10 mA cm^−2^ at a relatively low overpotential of 414.5 mV compared with >700 mV for both graphene and C_3_N_4_ and 510.2 mV for IrO_2_ [217]. The carbon film possesses a strong mechanical flexibility and very good structural properties, which justify its significant durability.

In almost all applications of the oxygen reduction reaction (but not the hydrogen peroxide production), it is desirable to select conditions where the complete 4-electron reduction reaction (ORR; O_2_ + 4H^+^ + 4e^−^ → 2H_2_O) occurs; i.e., in acid solution (pH 0, ORR formal potential +1.23 V vs. SHE) or in basic solution (pH 14, ORR formal potential +0.39 V vs. SHE). In any case, the oxygen electrode is a complex system and the overall reaction in either direction requires the transfer of four electrons and four protons. The majority of the research on the ORR has been centered on the use of noble metal electrodes, due to their relative stability in acidic or alkaline solutions. The preceding statement on noble metals for the OER does not hold true for the ORR. Using platinum as an example, we note that oxygen evolution is typically studied in the range 1.5–2.0 V vs. a reversible hydrogen electrode (RHE). In the same solution, oxygen reduction is studied in the range 1.0–0.4 V on the same scale. In most of the latter range, the surface is free of oxide (and very sensitive to impurities) if approached from low potentials, whereas it may be covered partially with oxide (and less sensitive to impurities) when approached from higher potentials. This is due to the high degree of irreversibility of formation and removal of the oxide layer on most noble metals. Again, considering statements above about the OER, it should be noted that the ORR may well occur on the bare metal surface, or one that is covered only by a fraction of a monolayer. Thus, even when Pt is used, it is well worth to remember that the oxidation and reduction of oxygen on the same metal occur at different surfaces and may therefore follow entirely different pathways.

More recently, emphasis has also been placed on the study of oxygen reduction on simple carbon electrodes, carbon electrodes containing electrocatalysts, non-noble single metal catalysts, single, binary and ternary metal oxide electrocatalysts, bifunctional unsupported and supported electrocatalysts, etc., in order to increase its rate and efficiency [218,219]. Generally, the ORR involves either four proton–electron transfers, or a two-proton–electron pathway, as discussed here and in the next section [220]. A direct four-electron mechanism can either be dissociative or associative [221]. An indirect four-electron mechanism involves first the two-electron pathway to hydrogen peroxide, followed by further reduction to water [22].
O_2_ + 2* → 2O*(32)
2O* + 2H^+^ → 2OH*(33)
2OH* + 2H^+^ + 2e^−^ → 2H_2_O + 2*(34)
O_2_ + * → O_2_*(35)
O_2_* + H^+^ + e^−^ → OOH*(36)
OOH*+H^+^+e^−^ → O*+ H_2_O(37)
O* + H^+^ + e^−^ → OH*(38)
OH* + H^+^ + e^−^ → H_2_O+*(39)

The free energies of all the above intermediates have been calculated on a variety of close packed metal surfaces, and a volcano plot was constructed relating the theoretical ORR activity and ΔE_o_, with Pt near the top. Certainly, it is desirable to promote the dissociative adsorption of oxygen to give M-O bonds of intermediate strength ensuring the appropriate free energy of adsorption. In addition, it may be advantageous for the oxygen molecule to be able initially to form a π complex at the surface and the final step may be assisted by polarization of the M-O bond, this being facilitated by using materials where the oxidation state may readily be altered. It should be noted that the O-O bond in oxygen is very strong and it is therefore not surprising that good electrocatalysts have proved difficult to find.

The synthesis of N-doped carbon-based nanomaterials for the ORR has been deeply studied, promising a prompt development of novel highly electroactive carbon nanomaterials for the ORR [222,223,224,225,226,227,228,229,230,231,232]. To further enhance the ORR performance, great efforts have been devoted, such as coordination with S and P heteroatoms [233,234,235,236,237], modification of carbon structure [238,239,240,241], and introduction of defects. The coordination of N with other heteroatoms can enhance the catalytic activity, which is ascribed to the synergistic effects. The catalyst with high specific surface area and porous structure that can expose abundant active sites due to ORR is an interfacial reaction. Thus, the design of nanomaterials with various nanostructure, such as 2D, 3D, and the coexistence of one-dimensional and two-dimensional structure, will provide sufficient active sites for the catalytic process. The introduction of defects in the carbon basal can change the distribution of electron density and electronic charge and then benefits the ORR process [242,243]. All the above strategies are not independent of each other. Thus, the combination of each strategy reasonable is also a significant method to optimize the catalytic activities.

Further and recent strategies to tune the pore size distribution and enhance the specific surface area also create abundant active sites for the ORR [244,245,246,247].

### 7.3. Hydrogen Peroxide Production

Hydrogen peroxide, well known for the last 200 years, is a very clean chemical with high versatility. It finds enormous application as a bleaching agent in the pulp, paper, and textile industries, as well as in the cosmetics and medicinal fields and the food processing industry. It is also used as an oxygen source and as an oxidizing agent in the mining and electronic industries. Being environmentally and ecologically friendly finds use in a variety of applications related to the environment. Highly concentrated hydrogen peroxide also finds use as a propellant iin the transportation industry.

The cathodic reduction of dissolved oxygen to peroxide was first demonstrated by Traube in 1882, but only over the last 50 years, there has been a gradual awareness of the desirable factors for a successful process, and cells developed by the Dow Chemical Company and others, lead to developed hydrogen peroxide production systems.

Interestingly, hydrogen peroxide is a carbon-free energy carrier that can be used as both fuel and oxidant in fuel cell engines (direct borohydride fuel cells, microfluidic fuel cells, direct alcohol fuel cells, on-chip fuel cells, and direct hydrazine fuel cells), having high energy density. The feedstock for electrogenerated hydrogen peroxide may be air (an unusually cheap and available feedstock) while its reactions lead only to oxygen and/or water [248]. Hydrogen peroxide is nontoxic, being more convenient and safer than compressed oxygen tanks.

Overall, the production of hydrogen peroxide from oxygen involves two coupled electron-proton transfers and one reaction intermediate (OOH*), making it similar in complexity to the HER [220].
O_2_ + * + H^+^ + e^−^ → OOH*(40)
OOH* + H^+^ + e^−^ → H_2_O_2_ + *(41)

As such, it is possible to find a catalyst with zero theoretical overpotential that has an optimal ΔG_OOH_, binding OOH* neither too strongly nor weakly [220]. Pt [249], Ag [250], Au [251], Au-Pd alloys [252], nitrogen-doped carbon [253], and hierarchically porous carbon [254] have been explored for the hydrogen peroxide production, but their activities and selectivities are very modest.

Figure 12A shows a volcano plot that relates the theoretical overpotential to ΔG_OOH_ for the two-electron reduction of oxygen to hydrogen peroxide [220], and experimental overpotentials at 1 mA cm^−2^ are overlaid on this plot [254]. As a result, the most promising catalyst with both high activity towards hydrogen peroxide would be found at the apex of the two-electron volcano plot. Now it is necessary to understand thermodynamics and kinetics, as well as interfacial processes of a broader range of materials and reactions to develop scalable catalysts highly selective to hydrogen peroxide while operating at low overpotentials.

Apart from the electroreduction of oxygen for hydrogen peroxide production, the electrochemical oxidation or reduction of hydrogen peroxide has wide practical application in sensors for medicine [255,256], food industry [257], and ecology [258]. The electrochemical reduction of H_2_O_2_ at low working potentials (usually around 0 V vs. Ag/AgCl), where interfering substances are electrochemically inert, allows for an optimum selective assay of the analyte. With respect to studies on the surface modified with micro- and nanosctructures carbonaceous electrodes as efficient electrocatalysts of hydrogen peroxide, both oxidation and reduction are currently on high demand [259,260,261,262,263,264,265,266,267,268,269].

### 7.4. Carbon Dioxide Reduction Reaction

The increased energy demand is gradually leading to the huge consumption of fossil fuels, and alleviation of the carbon dioxide effects caused by the heavy CO_2_ emission has become a pressing issue to the modern society. At present, the atmospheric concentrations of carbon dioxide are well over 400 ppm, and the rise in global average temperature and related negative environmental consequences (increasing death, asthma, hospitalization, cancer, etc.), is attributed to the increase in greenhouse gas emissions, particularly CO_2_.

The possibility of capture and storage of carbon dioxide in various media like amines, zeolites, and metal organic frameworks, as well as in geological systems, oceans, and by mineral carbonation has been technologically considered [270,271,272]. The capture and storage of carbon dioxide emissions can also be considered as a valuable resource because CO_2_ can be catalytically converted into industrially relevant chemicals and fuels [80,81,273]. A proposed method for the conversion of CO_2_ to value-added chemicals is the carbon dioxide reduction reaction (CRR), which was investigated on a wide variety of hererogeneous elemental surfaces, consisting of metal electrodes and other electrodes including metal complexes [274]. The CRR uses environmentally benign aqueous electrolytes, easily couples with electricity sources, and the reaction rate can be controlled easily by tuning the external bias (i.e., the overpotential). However, the currently known catalysts are very limited in terms of overpotential, selectivity, production rate, activity, and durability, hampering this process from becoming close to commercialization.

Linear carbon dioxide molecule is a non-polar, fully oxidized, and extremely stable molecule. To induce its chemical conversion, a commercial electrocatalyst requires a large overpotential to promote the conversion reaction at satisfactory rates and high product selectivity.

Figure 12B shows a volcano plot based on DFT calculations [274], that relates the theoretical limiting potential to DFT-calculated ΔE_co_ [275], and overlaid are measured onset potentials for the formation of methane and/or methanol, the earliest potentials at which either product is detected [276]. In the case of metals that bind CO* too strongly, the overpotential is dictated by the protonation of CO* to CHO*, whereas for metals that bind CO* too weakly, the overpotential is dictated by the protonation of CO (g) to CHO*, where CO desorption is the competing reaction [275]. For the formation of methane/methanol, Cu was found to reside near the top of the volcano plot with optimal ΔE_co_, albeit with significant theoretical overpotential of circus 0.8 V due to limitations from scaling relations [275,276]. Alternatively, carbon materials have been applied in carbon dioxide electroreduction with platinum catalysts supported on carbon nanomaterials with carbon chains >C5 [277,278]. Li et al. [279] addressed nanoporous S-doped and S, N-codoped carbons for the CRR of CO_2_ to CO and CH_4._

Carbon gels doped with transition metals have also shown activity in this CRR mechanism [280,281,282,283]. Regarding the hydrocarbon selectivity, recently [284,285], a high selectivity to C3-hydrocarbons among the detected products has been reported using CO- and Fe-carbon electrodes, which showed that the smaller the crystal size of iron, the higher the faradaic efficiency [285].

At present, the application of metal-carbon-carbon nanofibers composites obtained from plastic waste in the CRR to hydrocarbons is also being proposed for the transformation of LDPE polyethylene based plastic bags and other based residues in valuable products.

### 7.5. Nitrogen Reduction Reaction

Nitrogen is one of the most abundant elements in the atmosphere and most inert also. The electrochemical reduction of nitrogen (NRR) is more difficult than the CRR due to presence of three covalent bond between nitrogen atoms, which have high bond energy of 941 kJ/mol and therefore difficult to perform. For the NRR suitable catalysts are required, and Earth Abundant Electrocatalysts (EAEs) are good candidates for this reduction.

The most accepted mechanism of NRR contains associative and dissociative paths. Firstly, the nitrogen molecules are adsorbed on the catalyst surface and then the hydrogenation process proceeds. In addition, the hydrogenation process involves two approaches: Distal and alternating pathways. In the distal pathway, hydrogenation preferentially occurs on the N atom far away from the surface of catalyst. After formation and releasing of the first NH_3_ molecule, a second NH_3_ molecule is produced, while the hydrogenation alternates between two N molecules in the alternating processes. In the dissociative mechanism, the triple bond of N-N is broken first and then, the two N atoms adsorbed on the catalyst surface undergo a hybrid process independently. As with CRR, the NRR involves multiple intermediates, and the HER is a major competing reaction, making selectivity a great challenge [287].

As with CRR, the volcano plot in Figure 12C [286,288] shows a relation between the theoretical limiting potential to ΔE_N_ on a variety of metal surfaces. This plot again helps understanding the involvement of elemental metal catalysts for this reaction: metals that bind nitrogen too weakly are limited by the adsorption of N_2_ as N_2_H* in the first step of the reaction, while strong-binding metals are limited by either the protonation of NH* to form NH_2_* (flat surfaces) or the removal of NH_2_* as NH_3_ (stepped surfaces). Metals such as Ru, Rh, Mo, and Fe were calculated to lie near the top of neither the volcano plot, binding nitrogen neither too strongly nor weakly [288]. Unfortunately, even those metals that are found near the top of the volcano exhibit large theoretical overpotentials of at least 0.5 V due to non-ideal scaling relations between the intermediates. Thus, even these metals with the most active surfaces are predicted to be more active for promoting the HER rather than the NRR. At the right leg of the volcano figure, Rh, Ru, Ir, Co, Ni, and Pt prefer to absorb H-adatoms for the HER. Since the flat metal surfaces of early transition metals such as Sc, Y, Ti, and Zr bind N-adatoms more strongly than H-adatoms, significantly higher production of NH_3_ compared with that of H_2_ can be expected on these metal electrodes at the theoretical applied bias of −1.0 to −1.5 V. However, these metal surfaces are easily oxidized, thus it is hard to predict whether they are efficient NRR electrocatalysts or not. Modeling to this point suggests that a necessary condition for improved nitrogen reduction activity may be achieved by stabilizing N_2_H* relative to NH_2_* or NH* [286].

The electrochemical N_2_-to-NH_3_ process is currently being actively studied due to the importance of ammonia as fertilizer and in several plastics, fibers, explosives, pharmaceutical and textile industrial applications, as well as a high viable chemical energy carrier. The fundamental issue in much of the research on this electrochemical NRR process to form ammonia from nitrogen and water, is that the quantities of ammonia produced are very small. To minimize this problem, and in particular the simultaneous reduction of nitrogen and hydrogen, inhibiting the HER process, electrocatalysts are the paramount components. Thus, it is a priority to synthesize NRR electrocatalysts and rationally design them to optimize the mass transport, chemi (physi)sorption, and transfer of protons and electrons. Three important figures of merit are also being considered: The selectively of NRR over the HER, the energy efficiency of the overall process, and the system throughput of NH_3_ synthesis. Model systems in the nano-aqueous environment offer a guide to understanding the fundamental proton-coupled electron transfer (PCET) requirements needed to effect NRR over HER [289,290,291,292,293,294,295,296]. The parameter space for advanced catalytic molecules and materials remains to be explored. In this regard, computational studies help facilitate the catalyst design [288], while fine-tuning the active sites at nanoscale with chemical insight will be welcomed as a beneficial degree of freedom [297]. Conversely, hybrid biological inorganic (HBI) approaches [298,299,300] provide selectivity at ambient conditions and promise high-energy efficiencies, but with the challenge of sufficient throughput. Regardless of the recent use of sophisticated techniques and appropriate detection method protocols, the exact mechanism pathways on any given metal surface or flat and stepped surfaces (e.g., of noble metal, non-noble metal, and metal-free electrocatalysts) remains elusive for this hydrogen vs. ammonia issue.

### 7.6. Borohydride Oxidation Reaction

The Brown–Schlesinger and the Bayer processes were the first ways for producing NaBH_4_ [301,302,303]. Sodium borohydride is non-toxic, easily stored, has a high energy density (9.3 Wg^−1^), can be massively produced from trimethyl borate and borax, has long-term stability (half-life 426 days at pH 12), has potential usage at ambient conditions, has high hydrogen storage content (10.6 wt.%), releases up to eight electrons at an electrode potential as low as −1.24 V vs. SHE, and is pollution-free (the borohydide oxidation reaction, BOR, does not release carbon and nitrogen compounds, e.g., carbon dioxide, nitrogen dioxide, and so on). Apart from its oxidation, it has become the focus of many studies particularly in the field of the borohydride fuel cells [304]. The key factor for practical application of direct borohydride fuel cells (DBFCs) is to prepare anode electrodes (catalytic or not with respect to the borohydride hydrolysis) that have high selectivity and high catalytic activity for improving the kinetic parameters of BH_4_^−^ oxidation and inhibiting hydrolysis of BH_4_^−^. The ideal eight-electron BOR process can be understood as the oxidation of all the H(−1) in the borohydride anion into H(+1) in water. However, some of the H will be lost in the form of H_2_ during the oxidation, resulting in incomplete utilization of the eight electrons. The loss of faradaic efficiency due to the HER is a major issue limiting the performance of DBFCs. From the point of view of the cathode catalysts for the oxygen electrocatalysts, the alkaline electrolyte needed to stabilize the borohydride offers the possibility of using precious metals, non-PGM materials, activated and metal-doped carbons, nonmetal-doped carbon, carbon-transition metal hydrids, transition metal oxides with spinel and perovskite structures, and so forth. For alkaline fuel cells using carbon catalysts, the goal is to modify the cost-effective carbon-based electrocatalysts to increase the number of electrons up to four and to reduce the cathode activation overpotential. Pt-free ORR catalysts, the transition metal, nitrogen, and carbon groups, or M-N-C materials, are attractive candidates due to their high surface area, high activity, and low cost. The M-N-C synthesis involves various precursor deposition steps onto the high surface area carbons. The final catalysts consist of a combination of the active material with an inert carbon matrix, which substantially decreases the density of active sites for the four-electron pathway. Other recent ORR electroctalysts are being rationally designed in order to promote the electrocatalytic inner-sphere electron transfer by facilitation of direct adsorption of molecular oxygen on the active site, which enhances the activation of the peroxide intermediate on the active site that enables the complete four-electron transfer. In this context of advanced synthesis of ORR electrocatalysts, electrodeposition is a very simple and low-cost method to prepare such electrode materials without organic binder. The preparation of the materials in nanoscale with porous structure is also a useful way to obtain high efficiency with low quantity of catalyst.

Au alloys, Pt alloys, and other bimetallic materials, have been used as anodes for BH_4_^−^ oxidation [305,306,307,308,309] but poor long-term stability, corrosion, and other issues bare preventing their further use. Other materials, namely catalysts with core-shell structures on the other hand, have several unique and novel optical, electrical, and catalytic properties, and, as such, this type of catalysts has attracted increased research interest in recent years [310,311,312,313,314]. Recently, Duan et al. [315] prepared Ni@Au/C core-shell nanoparticles and first used them as anode electrocatalysts in the DBFC. The results showed that the DBFC using Ni_1_@Au_1_/C as an anode catalyst achieved a maximum power density of 74 mW cm^−2^ (at 130 mA cm^−2^), which was higher than those reported at Ni-Au/C alloy [316]. Among the transition metals, copper (Cu) has high activity for BH_4_^−^ adsorption and is considerably less expensive (more than 100 times cheaper) than Ag [317]. Thus, evaluating catalytic activities toward BH_4_^−^ oxidation of Cu@Ag core-shell electrocatalysts has merit. These carbon-supported bimetallic nanoparticles and other carbon nanomaterials [318] that are being synthesized and physically and electrochemically investigated are expected to be promising low-cost anode catalysts for DBFC application.

## 8. Direct Carbon Fuel Cell

Carbon materials and carbon nanomaterials are applied in many fuel cell technologies [319,320,321,322,323], which are being extensively explored. On the other hand, in the direct carbon fuel cell (DCFC), the overall investment is relatively small, and considerable effort is required to take this technology to the pre-commercialization stage. For this reason, and the fact that carbons, namely carbon anodes, play a key role in this system, we have decided to analyse this fuel cell here.

The DCFC uses carbon as solid fuel and oxygen as oxidant, both participating in the main electrode processes at the fuel cell. The cell operation does not need gasification, the carbon use can be almost 100% as fuel feed, the projected electrical efficiency can be above 75% (theoretically around 100%, i.e., a significant reduction in green house gas emissions), the amount of carbon dioxide for storage/sequestration is also greatly reduced, the exit gas is almost pure carbon dioxide stream, and the energy and costs to capture the carbon dioxide are expected to be much less than for other energy technologies. Furthermore, brown and black coal, coke, tar, biomass, and organic waste can serve as fuels. The problem is that there are many issues to be overcome, related to materials and corrosion, fuel delivery mechanism, reaction kinetics, and system development.

In a DCFC, each cell consists of a cathode and anode separated by an ionically conducting but electronically insulating electrolyte. Moreover, the anodic compartment is supplied with a solid fuel that reacts directly at the electrode/electrolyte interface or line of contact where the reactant, electronic conductor, and ionically conductive phase all meet, to form a gaseous exhaust product. These interfaces are known as triple-phase boundaries (TPBs). In the DCFC, the chemical energy in the carbon fuel (turned into submicron size particles) is directly converted into electricity, the overall cell reaction occurring at 500–900 °C:O_2_ (air) + 4e^−^ = 20^2−^(42)
C + 20^2−^ = CO_2_ + 4e^−^(43)

There are three basic types of DCFCs, depending on the nature of electrolyte used (molten hydroxide, molten carbonate, and oxygem ion conducting ceramic). There are also further sub-categories of DCFCs and those under development are summarized in Table 1.

Here, we describe in more detail a gasification-driven direct carbon solid oxide fuel cell (DC-SOFC), which is essentially similar to the conventional SOFC, enabling the use of solid carbonaceous fuels in SOFCs [324,325,326]. In these cells, the conventional Ni-YSZ (yttria-stabilized zirconia) cermet is the common anode (consisting of an electronically conductive and catalytic material) used in gasification-driven direct carbon SOFCs, as shown in Figure 13. There are three sub-categories of ECFCs that use an oxygenion conducting ceramic as the solid electrolyte. These differ only in the anodic compartment design and include fuel as solid carbon or in fluidized bed, fuel in molten metal, and fuel in molten carbonate (Table 1).

The key characteristics of DCFCs are:-Temperature of operation is currently 800–1000 °C;-OCVs obtained are in the range 0.95–1.10 V;-Power densities achieved of up to 150 mW/cm^2^ over a short testing period;-Electrical efficiency >80%;-CHP efficiency >90%;-Heat output 500–800 J;-High thermal insulation.

In the fuel cells developed by Gur and Huggins [324], the oxygen anions contained in the electrolyte react with CO in the porous anode forming CO_2_ and releasings electrons. Part of the CO_2_ is then recycled and subjected to a reverse Boudouard reaction, which is the gasification reaction between solid carbon and carbon dioxide to form carbon monoxide. The CO from the Boudouard gasifier is then cycled for further anodic oxidation.
CO + O^2−^ → CO_2_ + 2e^−^(44)
CO_2_ + C → 2CO(45)

The main issue here is that the reverse Boudouard reaction is an endothermic reaction and is kinetically very slow. Thus, it needs very high temperatures to occur and maintain the temperature of the gasifier [324]. Therefore, these indirect fuel cells possess low efficiency (about 55%), significantly lower than the potential >80% electric efficiency that may be offered by direct electrochemical reaction of carbon.

Mostly, the developmental work on this technology has so far been concentrated on button cells of ceramic electrolyte disk with nickel-based anode and lanthanum strontium manganite (LSM) based cathode [327,328]. The major technical issues apart from those associated with SOFC are the solid fuel delivery to anode/electrolyte interface, and the lack of understanding of carbon oxidation reaction mechanisms at the interface.

Many research groups are testing other single cells or small stacks, which possess low power densities, typically in the 100–120 mWcm^−2^ range and are strongly dependent on the fuel delivery system and the anode catalyst or current collector used. Table 2 below summarizes the technology status.

Two important factors have been identified as affecting the performance of molten-metal anodes for SOFC: The thermodynamic oxidation potential of the metal and the tendency for the oxide to form a film at the electrolyte interface, blocking the transfer of O^2−^ ions to the anode. For cells working in the battery mode, the oxidation determines the OCV that can be achieved. Low ionically conducting oxide films at the YSZ interface can effectively block charge transfer at the electrolyte interface. However, further work can be pursued to solve these two limitations. Apart from these factors, the following are also critical: (i) Wettability of carbon fuel in the case of molten media-based DCF; (ii) particle size, pore size distribution, and surface area; (iii) surface functional groups (nature and degree of functionalization); and (iv) type of impurities and their concentration.

Currently, most of the DCFC research in universities is directed at individual components of the fuel cell system using small button cells. Some research organisations (SARA, Contained Energy, and SRI International) are investigating complete systems and building stacks. However, a substantial effort is required to demonstrate DCFC technology in the kW range with reasonable life time (several thousand hours) with acceptable corrosion rates. In other words, a great investment in DCFC technology is urgently needed.

## 9. Neurochemical Monitoring

Microelectrodes, i.e., those with radii on the order of a few micrometers, have many unusual advantages over large electrodes, which are of interest to researchers in the field of sensor development. Because of the small size of microelectrodes, the absolute magnitude of the current obtained during an electrochemical experiment is quite small, usually a few hundred picoamps to several nanoamps. Currents of this magnitude are advantageous for the elimination of distortion in voltammetric response curves arising from the ohmic potential drop between the working and reference electrodes. This has led to the application of electrodes of this type to electroanalysis in non-polar organic solvents as well as in the absence of added supporting electrolyte. In situ monitoring in flow streams of changing ionic strength with microelectrodes, for example, should therefore be less prone to distortions in response due to changing resistivity of the bulk medium.

Because such low amounts of analyte can be electrolyzed at such a small electrode surface during an experiment, these electrodes are virtually non-destructive and have already found application as in vivo biosensors, most notably in the measurement of adrenergic neurotransmitters in mammalian brain tissue. The steady-state response typically observed at these electrodes at long times or moderate scan rates has been exploited for development of flow rate insensitive detectors for use in flow streams. Because more analyte is present at the interface than can ordinarily react with the electrode surface, vibrations or convection in solution have a minimal effect on the observed response. This small electrode size has the additional advantage of resulting in a greatly reduced capacitance and hence charging current, allowing for ultrafast voltammetric measurements with scan rates on the order of 10 billion V/s having been reported. Such fast-scan capability allows for both reduced analysis times and signal averaging of noisy signals.

More recently, it has been possible to design microelectrodes for biomedical applications of even smaller dimensions and lower electrical resistance in which the electronic conductor is a carbon fiber. Carbon fiber electrodes (CFEs) were first invented in the late 1970’s by François Gonon and colleagues [329,330], who discovered the carbon fiber electrode’s abilities to qualitatively detect species in the inner working of a brain by electrochemical pretreatment of the CFE before implantation [331]. This pretreatment helped to resolve oxidation currents measured in the rat brain into definite peaks scanned across the potential axis, enhancing CFEs in the use of selectively measuring electrochemical species in vitro and in vivo.

Highly oriented carbon fibers (graphite fibers) are characterized by extreme stiffness, exceeding that of steel. The typical diameter of commercial fiber is ca. 8 µm; this diameter may be reduced to less than 1 µm by air oxidation in a temperature gradient. It is known that carbon fibers can be modified by using intercalation methods or intensive oxidative surface treatment. The problem is to adjust the pretreatment procedure in such a way that the mechanical stability will not suffer. It is the aim of the pretreatment to attack not only the <edge> but also to open the <basal plane> carbon atoms, which provides an expanded electroactive surface. High current density anodic oxidation in aqueous electrolytes is the preferred method to produce thick layers of oxygen containing functional groups on carbon fibers. Repetitive oxidation/reduction current pulses promote the formation of uniform layers of oxides. These oxides have interesting properties: (i) In spite of the high degree of functionalization the oxides are still electronic conductors, because the anodic oxidation is automatically attenuated in areas where the electronic conductivity decreases due to overoxidation; (ii) the oxides are hydrophilic and microporous and can be soaked with any polar liquid; (iii) the oxides have surprisingly high cation exchange capacity (ca. 2 meq/g C). The combination of these three properties allows the insertion of the sensing material both on the surface of the carbon fiber and inside the carbon fiber structure. The advantage of CFEs over conventional potentiometric glass capillary microsensors is the noticeable lower internal resistance. Therfore, these low-impedance CFEs are being especially usefull for the monitorization of fast processes. It should be noted that the hydrophilic and microporous surface zone of carbon fibers can accommodate the reactive components of a typical reference electrode, e.g., Ag/AgCl/KCl(aq), as it has been reported in the open literature.

Previous preliminary studies showed that neurotransmitters and their metabolites were merely a percentage of key redox species present in the brain. In the 1980s, Mark Wightman and colleagues [332] determined guidelines to detect and identify in vivo species, which are now well known as the ‘Five Golden Rules. These golden rules are discussed in a book on voltammetry in the neurosciences [333] and the journal Trends in Analytical Chemistry [334].

Microelectrodes come in all shapes and sizes [109]. For the use in neurochemical monitoring, the two main kinds of shapes used for carbon fiber electrodes are the needle and disk electrodes. Typical dimensions of the carbon fiber tip range from 7–20 µm in diameter and 400–800 µm in length. The order of magnitude of the area of a CFE is in the 10^−4^ cm^2^ range. Since the electrode is directly implanted into the brain, any signal recorded has only a time delay of the length of time it takes the analyte (e.g., dopamine) to diffuse across a gap approximately 3–4 µm to the electrode, which is on the order of milliseconds [335,336]. In addition, CFEs can be used to measure cellular and subcellular events easily. These kinds of investigations are unrealistic and incapable for larger measuring devices (Figure 14) [337].

The small size CFEs implanted in the brain for neurochemical monitoring can also be used for quantal size and vesicular volume studies [338], and for the detection of a wide range of species in the brain, with a small impact in detrimental damage to the surrounding tissue near the implantation site than much larger devices. Two relevant neurotransmitters in the brain are dopamines and ascorbic acid, as well as their metabolites. In vivo, or in physiological media, these species become oxidized and release electrons, and have been studied with small size microelectrodes and voltammetry [339].

The ability to distinguish between oxidation signals arising from catecholamines and ascorbic acid is an analytical problem of considerable interest to bioelectrochemists. In vivo electrochemical measurements for the determination of adrenergic neurotransmitters are complicated by selectivity problems arising from the similar redox potentials of catecholamines and the interferent ascorbic acid, the latter of which is presnt in mammalian brain tissue at concentrations roughly an order of magnitude higher than the combined levels of dopamine and its metabolite, 3,4-di-hydroxyphenylacetic acid (DOPAC) (ca. 280 µm vs. 25 µm). At an unmodified CFB, the oxidation peaks from square wave voltammetry for a solution 1 mM in dopamine and 10 mM in ascorbic acid at pH 7 are totally unresolved [340,341].

Various strategies have been adopted for modifying carbon surfaces in order to impart the necessary selectivity for analyses of this type. Most of these methods rely on the introduction of negatively charged surface species to repel both ascorbate and uric acid (each are deprotonated at physiologic pH), while the signal for dopamine, which is protonated at this pH, is not attenuated. The employment of coatings of the anionic, sulfonated fluoropolymer Nafion has been developed for the above purpose, but the results were similar. Moreover, this coating approach has the distinct disadvantage of also discriminating against the anionic dopamine metabolite DOPAC which, in addition to often being the catecholamine present in highest concentration in biological fluids, is itself an analyte of interest. More recent work has shown that the nature of this phenomenon has been identified as a specific polymer/cosolute interaction arising from hydrogen bonds formed between amide centers along the polymer chain and the phenolic-OH groups present in all catecholamines. The basis of the selectivity exhibited by these electrodes is thus fundamentally different from that of the other devices described above. The selectivity does not depend on the charge of the analyte, but instead relies on an interaction with a structural feature of catecholamines not present in the two most important interferents (ascorbate and uric acid), namely an aromatic hydroxyl group. In summary, specifically, the response of modified electrodes to dopamine in the presence of ascorbic acid present at a concentration 10 times higher is enhanced considerably. The possibility of developing electrodes that respond to catecholamines of different charges exists, but not to non-aromatic interferents since the nature of the selectivity of these electrodes is not based on a simple coulombic repulsion scheme.

Electrochemical methods are also used to detect small molecules and biologically significant ions. Ion selective electrodes are quite often used in vitro; however, they can be miniaturized for in vivo use for various applications, such as the pH electrode [340], the oxygen sensor electrode [341], the nitric oxide sensor [337], and the hydrogen peroxide enzyme electrode [342]. These enzyme-based microsensors, namely the oxidase enzymes that are nonelectroactive [343,344], particularly used for the detection of choline, glutamate, glucose, and lactate, are commonly used.

Typically, the enzyme gets oxidized in a reaction with a co-substrate [345], that usually is oxygen (O_2_), which generates hydrogen peroxide (H_2_O_2_) as the final co-product. Hydrogen peroxide can be electrochemically oxidized and detected on the electrode surface, its rate of production ideally being directly related to the concentration of the substrate detected by enzymes.

Future directions for the carbon microelectrode are likely related to its size. At present, carbon nanotubes (CNTs) are the point of interest [346,347,348]. Apart from their use in neurochemical monitoring, carbon fibers and carbon nanotubes, either single-walled SWCNTs or multi-walled MWCNTs [349,350,351], are finding their way in different applications that touch nearly every field of technology including aerospace [352], electronics [353], medicine [354], defense [355], automotive [355], energy [356], construction [357], and even fashion [358]. In particular, the ability of CNTs to effectively target the cancer cells may revolutionize our approaches in treating this dreaded disease and bring us near the Holy Grail [359,360,361,362,363,364,365].

## 10. Lithium Ion Batteries

Energy storage has long been recognized as a meaning of reducing petroleum demand and alleviating air pollution problems. Increasing public concerns about air quality have led to new requirements for zero-emission and low-emission vehicles in Europe, Japan, United States, and elsewhere in the world by the last two decades. Advanced secondary lithium-ion batteries are being developed to provide electric vehicles (EVs) with the range, acceleration, and low life-cycle costs necessary to penetrate commercial markets. These batteries are also ideal for electric utility load leveling applications, providing fuel flexibility and reduced atmospheric emissions, as well as offering valuable operating benefits (e.g., spinning reserve) to electric utilities [366].

The four most-used types of comercial rechargeable batteries are lead-acid [367], nickel-cadmium [368], nickel-metal hydride [369], and lithium-ion [370]. The lead- acid battery, which was introduced by G. Planté in 1850, is the most widely used rechargeable electrochemical device. Recent developments include (i) optimized electrode material formulations; (ii) novel electrode structures and current-collector designs; (iii) better electrode materials to reduce corrosion problems and improve electrode cohesion; (iv) cell designs affording electrolyte circulation; and (v) immobilized electrolyte (the electrolyte may be gelled or absorbed in a glass mat, etc.) cell designs. The nickel-cadmium battery, which was introduced by W. Jungner in 1899, has high-rate and low-temperature performances, better than those of lead-acid, and other beneficial features are: A flat discharge voltage, long life, continuous overcharge capability, and good reliability. The nickel-metal hydride battery, which was commercialized in 1989, has an operating cell voltage of 1.2–1.3 V, which is almost the same as that of nickel cadmium, allowing ready interchangeability, and the discharge curve is quite flat. Moreover, nickel-metal hydride batteries are capable of producing pulses of very high power; they are also resilient to both overcharge and overdischarge, and may be operated from −30 to +45 C. Another attraction is that there are no toxicity problems with recycling. The lithium-ion battery was commercialized in 1990.

The lithium ion battery (LIB) technology is of unequivocal dominance over the other batteries [371], because LIBs have high energy density and high cycle number; that is, long life. However, the price of cobalt and lithium is increasing, and so is interest in new chemistries [372,373]. Sodium-ion batteries and those based on calcium or magnesium use materials that are more plentiful [374]. They have potential for electric vehicles and for utility energy storage, which will both become huge markets as the use of fossil fuels is reduced. In fact, the energy storage market has the potential to become the largest market for batteries so that energy from renewables, solar, and wind in particular can be stored for later use [375]. This demands much lower cost energy storage and both new battery chemistry and continuing development of existing technologies for lithium-, lead-, and zinc-based batteries will allow this market to be developed further [376]. Lithium-air, zinc-air, lithium-sulphur and sodium-sulphur batteries are promising next-generation energy storage technologies with projected specific energies up to 600 Wh/kg and over on cell level, but their complex multicomponent devices are affecting their performance, so their deeper mechanistic understanding is required to determine lifetime, cyclability and capacity of the whole devices before moving them to further commercialization [377,378,379]. Therefore, in this section a focus is on the carbon nanomaterials for the lithium-ion system as the most promising technology to date [380,381,382].

Lithium batteries were first proposed, by M. S. Whittingham around the 1970s [383]. The group IV and V dichalcogenides, later identified as intercalation compounds, attracted attention for their high electrical conductivities, and the fact that they react with alkali metals in a reversible way. In 1972, Exxon [384] initiated a large project on solid state batteries, using TiS_2_ as the positive electrode, Li metal as the negative electrode and lithium perchlorate in dioxolane as the electrolyte. Titanium disulfide was chosen as a cathode material because it is the lightest and cheapest of all groups IV and V layered dichalcogenides and its ability to undergo intercalation upon treatment with electropositive elements. TiS_2_ adopts a hexagonal close packed structure, each sulfide being connected to three Ti centers, the geometry at S being pyramidal [385]. The individual layers of TiS_2,_ which consist of Ti-S bonds, are bounded together by relatively weak intermolecular van der Waals forces. Unfortunately, it was soon realized that using lithium as an anode material lower the performance of the batteries (low capacity and cycle life) and made them unsafe due to dendritic Li growth during charge-discharge cycling. These dendrites or pesky formations are the reasons that most lithium batteries on the market today are lithium-ion rather than lithium-metal.

The 1979 discovery at Oxford University that Li^+^ ions may be electrochemically withdrawn from the LiCoO_2_ and LiNiO_2_ structures and replaced reversibly, that is, either of these compounds can be used as the active material for a positive electrode in a 4 V rechargeable lithium cell, was a significant advance. In 1990, the Sony Corporation in Japan, announced the <lithium-ion battery>, that was the first rechargeable lithium battery depending entirely on the difference in electrochemical potential of lithium ions intercalated in LiCoO_2_ and graphite.

To obtain high energy density batteries, graphite, graphitizable carbon (soft carbon), and nongraphitizable carbon (hard carbon) [386], with large doping capacities, and the possibility of lithium-carbon intercalation complexes exceeding the LiC_6_ stoichiometric composition, are being studied and employed as anodes.

Graphite has been generally used as anode due to its large capacity, and excellent cycle life. Recently, non-graphitizable carbon materials have been studied as alternatives for graphite due to the interlayer distances, which are large compared with those in graphite and soft carbon, and their high working potential, which could avoid Li electrodeposition during charging.

However, non-graphitizable carbon also has some shortcomings: A large difference in potential is seen between charge and discharge reactions, and a large irreversible capacity. In 1980, Rachid Yazami demonstrated the reversible electrochemical intercalation of lithium in graphite: By using a solid electrolyte based on polyethylene oxide and lithium perchlorate (formula P(OE)_8_–LiClO_4_), he reported the world’s first successful experiment demonstrating the electrochemical intercalation and release of lithium in graphite [387,388].

The fact that lithium ions can be inserted/deinserted (or intercalated/deintercalated) in graphite is due to the strength disparity in the forces between any two given carbons in the same sheet (which share sp^2^ hybridized bonds) and the forces between any two concurrent sheets [389]. This storage density [390], often called capacity, similarly has a theoretical limit; in the case of graphite, it is 372 milliamp hours per gram (mAh/g) [389], which is a relatively low capacity. Furthermore, the graphite’s low expansion is directly linked to their ability to maintain their charge capacity after many charge-discharge cycles. Their predominance in the market is a result of their cycle over cycle efficiency, not their capacity.

The development of lithium cobalt oxide, lithium manganese oxide, lithium iron phosphate, and other positive electrode materials [391,392,393], the development of solid electrolytes that also act as separators [394], and other separators with very good electronic insulation and the capability of conducting ions by either intrinsic ionic conduction or by electrolyte soaking [395], are minimizing the processes that adversely affect the electrochemical energy efficiency of the batteries.

In the 1980s, Yoshino developed a new nonaqueous electrolyte secondary battery to meet the emerging need for a small and lightweight power source for portable electronics. He completed a practical prototype in 1986, using carbonaceous materials with a certain crystalline structure as the negative electrode, and LiCoO_2_ as the positive electrode [396,397,398,399,400,401].

This combination achieved an electromotive force of 4 V or more, provided stable battery characteristics over a long service life, including excellent cycle durability with little degradation by side reactions and excellent storage characteristics [397]. Yoshino also developed novel technology for fabricating electrodes and assembling batteries, namely for preventing ignition [402]. A multilayer electrode assembly (electrode coil), is inserted into a battery can, that is infused with nonaqueous electrolyte of LiPF_6_ or LiPF_4_ dissolved in a mixture of carbonate compounds, and sealed [397,403]. Another approach was the development of peripheral technology, including safety device, protective circuit, and charging and discharging technologies.

These achievements by Yoshino led Sony to release the world’s first commercial lithium ion rechargeable battery product in 1991 [404]. The first LIB anode electrode was lithium metal but soon it was replaced by graphitic carbon and other carbon materials with large doping capacities, and the possibility of lithium carbon intercalation complexes exceeding the LiC_6_ stoichiometric composition.

The next generation of anode materials included nanomaterials whose use led to high capacity life LIB negative electrodes. This electrochemical intercalation ability of LIBs explains the benefits of using nanomaterials and nanotechnology to improve their performances, as well as the fabrication of small micro and nano batteries with faster charge-discharge reactions, new mechanisms for lithium-ion storage [405,406], enhanced mechanical properties [407], and so on.

Recently, metal oxides nanoparticles encapsulated by graphene layers [408] have been reported to display high specific capacity and excellent cycling performance. The graphene layers actuation both as a ‘buffer zone’ of volume variation of the nanoparticles and a good electron transfer medium, was believed to be the key issue.

Graphene is expected to be a good electrode material, due to its intrinsically superior electrical conductivity (>100 S cm^−1^), excellent mechanical flexibility, remarkable thermal conductivity (around 3000 W mK^−1^), and high surface area, as well as the open and flexible porous structure of graphene powders. The high chemical diffusivity of Li (10^−7^ to 10^−10^) cm^2^ s^−1^, on a graphene plane also contributes to its high-power application. Currently only chemical exfoliation of graphite is potentially capable of the large-scale production of graphene to meet the requirements for LIB applications [409]. A problem with graphene is its structural re-stacking nature [410], which required its integration into a hybrid material to obtain a synergy effect. With the use of graphene, some morphologically modified carbon nanostructures such as carbon nanotubes (CNTs) [411], fullerenes, activated carbon [412], and carbon aerogels [413] have been reported.

One great challenge in the development of lithium ion batteries is to simultaneously achieve high power and large energy capacity at fast charge and discharge rates for several minutes to seconds [414]. In this aspect, transition metal oxides, silicon, tin, and zinc, etc., with addition of additives to mitigate volume changes observed during cycling have been explored as active anode materials to replace graphite because of their high theoretical capacities.

In recent years, graphene has been employed as an encapsulating agent for these materials. So, this work that eventually provides materials with high capacities requires consideration. Furthermore, stacked sheets of graphene derived from exfoliated graphite provide a modular approach to exploring lithium storage in layered carbon as well as layered carbon/metal nanocomposite [415].

Owing to its high theoretical capacity (4200 mAh/g) when alloying with lithium, and the fact that is the second most abundant element on earth, silicon has received a huge attention as a prospective replacement material for use as anodes in a LIB [60,416] though it is actually a metal. The alloying of silicon with lithium is associated with a volume expansion of more than 300%, leading to pulverization, which results in loss of electrical contact and eventual fading of capacity. Enormous efforts have been made to overcome these problems by using nanostructured Si materials and others, but it remains challenging via facile approaches to achieve long cycle life and high capacity of Si anode materials in large scale [417,418,419,420,421].

A mechanically encapsulated silicon graphene composite was prepared by Dou et al. [422] by simple mixing of commercially available nanosize Si (40 nm) and graphene in a weight ratio of 1:1 by mortar. The Si/graphene composite electrode showed calculated contribution of the pure Si. However, the aggregation of Si nanoparticles and electrolyte was inevitable and capacity fading was caused. There were many attempts to solve the aggregation problem [423,424,425,426], and silicon expansion was relieved to some extent and nanoparticles aggregation was further minimized. A more effective method consists of encapsulating a metal oxide into a graphene layer by co-assembling between negatively charged graphene oxide and positively charged oxide nanoparticles [427]. Hwang et al. [428] reported a novel Si/G composite by non-covalent anchoring of Si nanoparticles onto the surface of graphene sheets by electrostatic attraction followed by thermal processing to remove residual organic material.

Further studies [429,430,431] led to three-dimensional (3D) silicon/carbon/graphene nano composites, three-dimensional (3D) graphene-carbon nanotube-metal/metal oxide nanocomposites, three-dimensional (3D) graphene-carbon nanotube–TiO_2_ nanocomposites, three-dimensional (3D) graphene-carbon nanotube-nickel nanocomposites, and other novel 3D functional nanostructures, namely three-dimensional (3D) graphene-nanotube-iron hierarchical nanocomposites and doped hierarchical porous graphene electrodes [432,433,434,435] with outstanding structural properties, minimum contact resistance, enhanced Li-ion diffusion, high capacity, high energy density, and prevention of the restacking of graphene layers. In addition, these structures overcome the agglomeration of metal/metal oxides nanoparticles in the case of 3D composites.

## 11. Electrochemical Capacitors

Renewable energies, that is, natural energy resources such as wind power, solar energy, and geothermal energy, are converted to usable energy, mainly electric energy, which is stored in batteries and capacitors, particularly of the type’s lithium-ion and electrochemical capacitor. LIBs were analysed in the previous section. Here, we deal essentially with electrochemical double layer capacitors (EDLCs). These batteries and capacitors utilize carbon materials as electrodes.

A conventional electrostatic capacitor consists of two metal plates of equal area, A, separated by an insulator (a dielectric, e.g., vacuum, air, mica, oil, paper, plastic). By means of an applied voltage, the device stores energy by the separation of positive and negative electrostatic charge, q, between the parallel plates. If the potential difference between the plates is V_c_ and the dielectric constant of the insulating material is ©, then the capacitance, C, is given by C = q/V_c_ = © A/d, where d is the inter-plate spacing. The energy stored in a charge capacitor is given by U = ½ C V_c_^2^. The energy density is very low, of the order of 0.05 Wh dm^−3^.

In an electrolytic capacitor, the dielectric is a very thin oxide film formed electrolytically on a metal such as Al, Ta, Ti, or Nb. The metal substrate acts as one conducting phase and an electrolyte solution as the other. Given that the dielectric is extremely thin, the specific capacitances (F/g) of electrolytic capacitors are much larger (up to 1000 times) than those of electrostatic counterparts. The specific energy is higher (typically 0.06 vs. 0.003 Wh/kg), but both types of capacitor are extremely poor energy-storage devices.

So-called supercapacitors store electrostatic charge in the form of ions, rather than electrons, on the surfaces of materials with high specific areas (m^2^/g). The electrodes are of finely divided, porous carbon, which provide a great charge density. The voltage is lower than for a conventional capacitor, while the time for charge-discharge is longer because ions move and reorientate more slowly than electrons.

By combining an electrode material with a large specific surface area with a material that can be reversibly oxidized and reduced over a wide potential range, it is possible to realise a device close to a battery, that is the ultracapacitor. The energy is stored both by ionic capacitance and by surface (and near surface) redox processes that occur during charge and discharge. This enhances the amount of stored energy. Moreover, because the ions are confined to surface layers, the redox reactions are rapid and are fully reversible many thousands of times, which therefore make for a long lifecycle.

Electrochemical capacitors (ECs) can store vastly more energy than conventional capacitors. They vary in size from small capacitors used in electronics to devices with capacities >3000 F that form the basic module for the units used in hybrid electric vehicles. They may be discharged at rates up to 10 to 25 times faster than batteries and, equally importantly, can also be recharged at much greater rates than batteries. Moreover, they have very long lives and operate satisfactorily at temperatures as low as −40 °C.

Electrical energy storage in ECs occurs due to the formation of electric double-layer (EDL) on the electrodes’ surfaces and to some extent surface oxidation/reduction. The capacitance due to the former (EDLC) is called electric double-layer capacitance and that due to the latter is called pseudo-capacitante. The capacitors that consist of different mechanisms, for example, intercalation/deintercalation and adsorption/desorption, are called hybrid capacitors. Activated carbons with a large surface area, good electric conductivity, electrochemical inertness, and lightweight properties, are excellent materials to increase the capacitance, so that they are very employed as carbon electrodes. Many reviews and articles on carbon materials to EDLCs have been published [436,437,438,439,440].

Limitations of space only allow a very brief summary of a few carbon types such as activated carbons, templated porous carbons, and others, for use as electrodes, mainly in EDLCs. Nanoporous carbons prepared by different methods include activated carbons (ACs) [441,442,443,444], activated carbon fibers (ACEs) [445,446,447,448], exfoliated carbon fibers [449,450,451], templated porous carbons [452,453,454,455], polytetrafluoroethylene (PTFE)-derived carbons [456,457], carbide-derived carbons [458,459,460,461], and carbon aerogels and xerogels [462,463]. Some of these materials presented high surface area and different EDLC capacitances, which could be explained by different contributions from microporous and external surfaces, and the introduction of mesoporosity, with many changes in the preparation conditions. Carbon nanotubes (CNTs) with large area of exposed surface and different storage spaces for electrolyte ions, apart from their high electrical conductivity, are also expected to be very attractive for capacitor electrode materials. Capacitive performance of simple-walled carbon nanotubes (SWCNTs) have been measured by using various organic and aqueous electrolytes [464,465,466,467], and many of them show more excellent properties as electrode materials than ACs.

The capacitor properties of double-walled carbon nanotubes (DWCNTs) have not been so frequently reported, but commercially available SW and DW (HiP-coTM) [468] showed similar electrochemical properties. Larger capacitance values have been reported for MWCNTs, which are relatively easy to be synthesized. Depending on the synthesis methods and the modification, their capacitance values are widely distributed in both aqueous and non-aqueous electrolytes from 10 to 200 F g^−1^ [464,469,470,471]. It has already been reported that carbon materials can have various functional groups on their surface. Some of these functional groups contribute to the capacitance of EDLCs, which is called pseudo-capacitance [472]. More specifically, pseudo capacitance corresponds to rectangular cyclic voltammograms (CVs), and linear plots of galvanostatic charging and discharging (GCDs). In the case of oxygen-containing functional groups, pseudo-capacitance is generally credited with faradaic reactions of these groups with electrolyte ions, which is basically identical with the capacitance developed by transition metal oxides, such as RuO_2_ and MnO_2_ [472,473,474,475]. Note that the electrode materials in batteries also rely on the faradaic charge storage mechanism, which is, however, non-capacitive, corresponding to peak-shaped CV and non-linear GCD.

For most of the porous carbons, the ratio of Cg to S_BET_ (Fm^−2^) is hard to attain 0.2 Fm^−2^. With carbon materials containing nitrogen at certain levels, Cg/S_BET_ readily exceeds this range and becomes more than 10 times of it in the extreme case [476]. Increasing capacitance by nitrogen doping is often attributed to faradaic reactions of the nitrogen-containing functional groups (e.g., [477,478,479]), analogous to the case of oxygen-containing carbons. At present, it is difficult to distinguish and determine other contributions as well as to explain the effect of nitrogen doping reasonably [480,481,482,483,484,485,486,487,488,489].

Concerning the pseudo-capacitance by boron-doping into carbon material, only a limited number of references are found [490,491,492]. However, boron and nitrogen co-doped carbon materials were reported as good for electrochemical capacitors [493,494,495,496,497,498,499]. Recent capacitors constructed by using a combination of a microporous with different mesoporous carbons were shown to be good as asymmetric EDLCs [500,501]. Hybrid capacitors consisting of different storage mechanisms have also been proposed [502,503,504,505,506].

In very recent years, supercapacitors have gained prime importance due to their escalated power density, speedy charging/discharging, long cyclic efficiency, ecological suitability, and cheaper value [507,508,509,510]. At this point, it should be noted that EDLCs, hybrid capacitors, and pseudocapacitors are the three types of supercapacitors that are being largely discussed from both research and application perspectives, in the context of electrochemical energy storage (EES) devices. Table 3 compares supercapacitors with capacitors and batteries. However, the realization of supercapacitors in various applications is still hindered by its low energy density compared to conventional batteries [511]. Table 4 compares the main differences in the properties of batteries and supercapacitors. Therefore, investigations are carried out to increase the energy density of supercapacitors without compromising their power density. To summarize, ECs are entirely complementary to batteries, but obviously they would be employed only in situations that require transient pulses of high power.

Recently, hybrid energy storage devices termed as supercapattery have been developed to complement the features of supercapacitors and batteries. The supercapattery uses redox active battery-grade materials as positive electrode with the high power-delivery capability and carbonaceous materials as negative electrode [512]. Thus, complementing the advantages of both batteries as well as supercapacitors, i.e., possessing energy as much as the battery and high power output almost as much as the supercapacitor.

At the materials level, all highly porous structures and nanoparticulates or redox active materials, either capacitive or non-capacitive, can be used for supercapattery. In addition, pseudocapacitive materials represent a special case of supercapattery electrode materials because they are both faradaic and capacitive in nature. In comparison with other terminologies for hybrid ESS devices, supercapattery can provide a common conceptual basis for analysis, comparison, and communication. For a hypothetical supercapattery consisting of a Li metal negative electrode (negatrode) and a 400 F/g supercapacitor positive electrode (positrode), a higher energy capacity than the Li ion battery is expected. In fact, since the specific charge capacity of Li is much larger than that of the supercapacitor electrode, the Li mass is negligible in the supercapattery. Then, the theoretical specific energy for discharging the cell from 3.5 V to 1.0 V would be 400 × [(3.5 × 3.5) − (1.0 × 1.0)]/(2 × 3.6) = 625 Wh/kg. Reviews on supercapattery technologies have been published recently, and it is believed that they would promote the performance of present EES devices in terms of energy capacity, power capability and cycle life [513,514,515].

## 12. Conclusions

About 15,000 years ago, petroleum and coal appeared, making life very easy for modern man. The waste products of the petroleum industry (the distillation of the barrel), and the waste products of the coal carbonization industry (the manufacture of metallurgical coke) as coal-tar pitch, are used to create the carbon artefacts (matrix and binder) of the carbon electrode industry.

During the first half of the last century, it had become obvious that the route to aluminium production was via the Hall-Héroult cell, i.e., the electrochemical reduction of alumina, by carbon, in a molten bath of cryolite. Developments of the carbon anode had pointed the way to the use of a coke bonded with coal-tar pitch. At the same time as the aluminium industry was expanding, the petroleum and steel making industries were providing the necessary ingredients of anode manufacture. Clearly, this exploitation and hence continuous quality control of coke and pitch for the anode is a necessity. Carbon blacks associated with printing inks are an essential ingredient of the automobile tyre of plastics in the modern car, and in the paint that covers the modern car. Aircraft use braking systems of carbon composite made up of carbon fiber matrices bonded with carbon from coal-tar pitch. Also, the steel industry makes its steels in the furnace heated using the graphite electrode, made from premium-quality delayed coke (needle coke) and coal-tar pitch. In the present century, sophisticated carbon electrodes have been manufactured for many applications, namely in the area of electrochemical energy devices [516,517,518,519,520,521,522,523,524,525].

Materials science and engineering meetings, as well as many others involving several industries using carbon electrodes, continue to show that there are many factors that need to be considered and improved to obtain efficient anodes/electodes. Restricting to the aluminium electrowinning we really do not understand very well what happens within the green anode when we pyrolyse and bake it, namely we do not know exactly how to modulate optimum relationships between coke particle (shape and size), butt particle, and mixing extent with coke particles, and the shape and size of the binder coke bridges.

Looking back at the carbon highlights reported here, we clearly found some areas deserving attention. It is the case of the aluminium production in large alumina refineries, using carbon anodes of high quality, which depends on the characteristics of coke filler, coal tar pitch binder, and anode scrap, among others. However, then we can see the development of synthetic diamonds by the GE high pressure catalytic process initiated in 1941, leading to the first commercially successful synthesis on december 1954. Much later the diamond and diamond-like films appeared, using low temperatures and low-pressure procedures, truly defiant of all the laws of thermodynamics and phase diagrams. The carbon fibers are other excellent carbon materials whose development led to the carbon fiber reinforced plastic (CFRP) and other composite products, which have several uses in aerospace and non-aerospace structures, as well as in non-structural applications. One of the most intriguing discovery has been the fullerene systems and the nanotubes that are capturing the imagination of physicists, chemists, materials scientists, and nanotechnologists alike. These new discoveries and developments had an impact that extended well beyond the confines of academic research and marked the beginning of a new era in carbon science and technology, namely in the field of carbon electrochemistry.

Another factor stimulating the research interest on carbon for electrocatalysis is the worldwide need to develop more efficient approaches for electrode materials for more sustainable utilization of energy together with the possibility of fine tuning of their nanostructure to realize advanced electrodes to meet the demanding expectation for more sustainable and efficient conversion and storage of energy.

For a better understanding of these carbon-based electrodes development and applications, this article includes a brief account of structure in carbons and carbon forms, followed by catalysis of carbon oxidation reactions, nanotechnology, and carbon electrocatalysis. The traditional aluminium electrolysis is then reported, followed by the application of carbon materials in electrochemical kinetics and new electrochemical energy technologies, namely direct carbon fuel cells, lithium ion batteries, and electrochemical capacitors. Updated applications such as neurochemical monitoring and supercapattery are also reported.

## Figures and Tables

**Figure 1 molecules-25-04996-f001:**
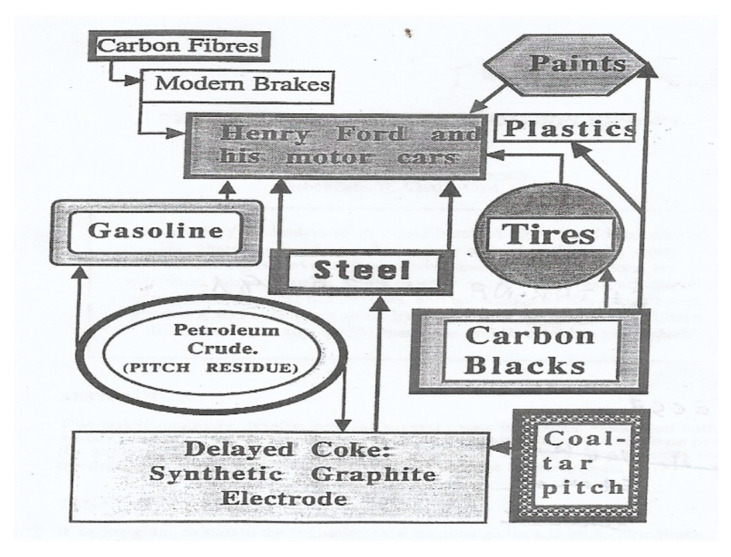
A schematic illustrating the interdependence of carbon industries.

**Figure 2 molecules-25-04996-f002:**
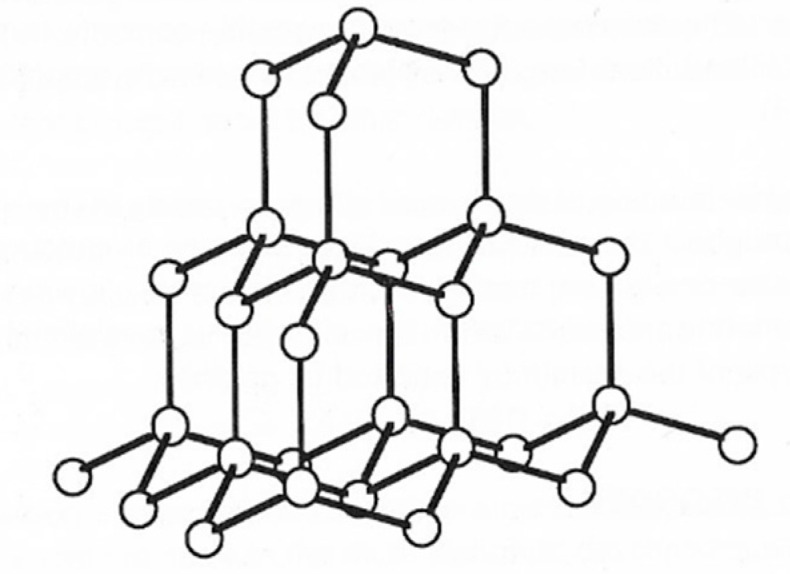
Structure of diamond.

**Figure 3 molecules-25-04996-f003:**
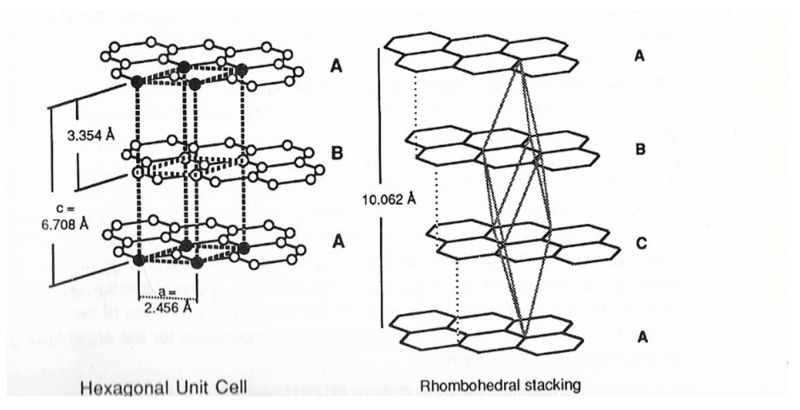
Graphite structure.

**Figure 4 molecules-25-04996-f004:**
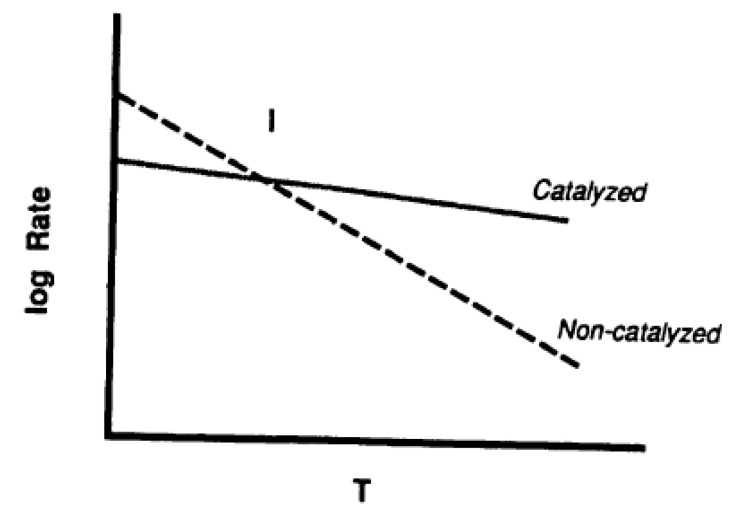
Arrhenius plots of catalyzed and non catalyzed gasification showing an isokinetic point I.

**Figure 5 molecules-25-04996-f005:**
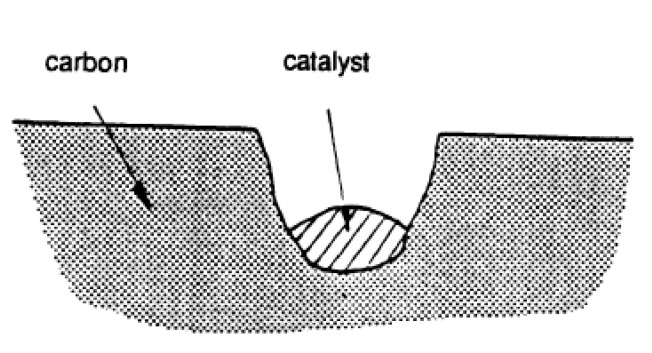
Pitting of a graphite basaf plane.

**Figure 6 molecules-25-04996-f006:**
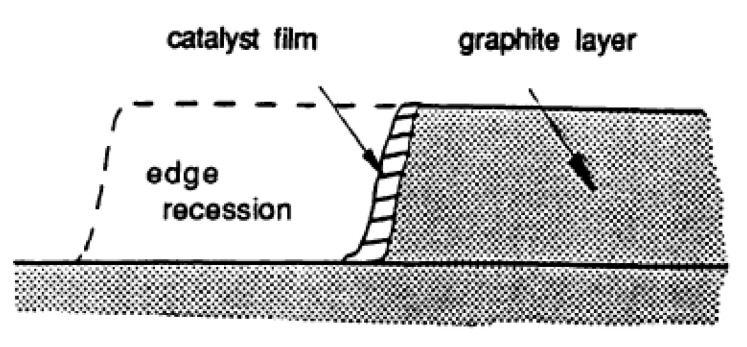
Catalytic edge recession of a graphite basal plane.

**Figure 7 molecules-25-04996-f007:**
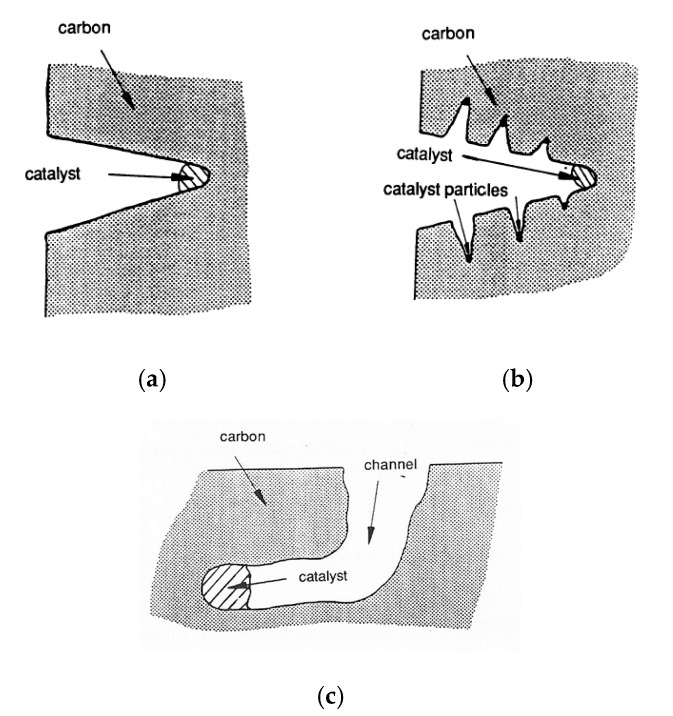
(**a**,**b**) Catalytic channeling action resulting in fluted channels. (**c**) Changing direction of a channeling catalyst particle.

**Figure 8 molecules-25-04996-f008:**
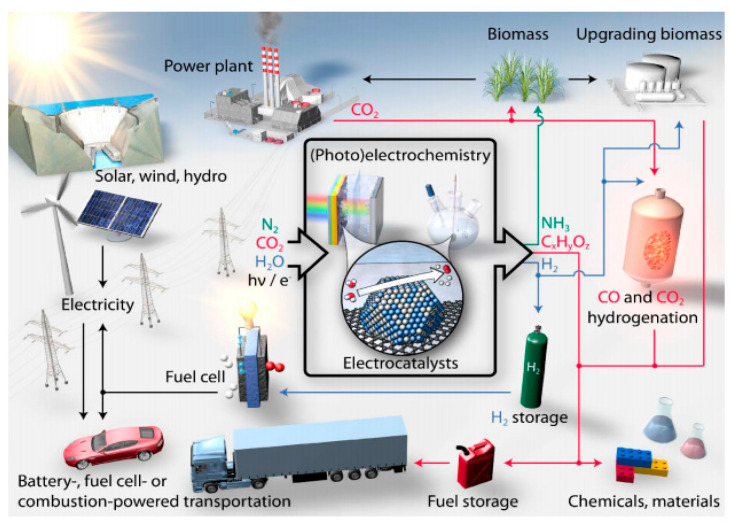
Sustainable energy future. Schematic of a sustainable energy landscape for the future based on electrocatalysis.

**Figure 9 molecules-25-04996-f009:**
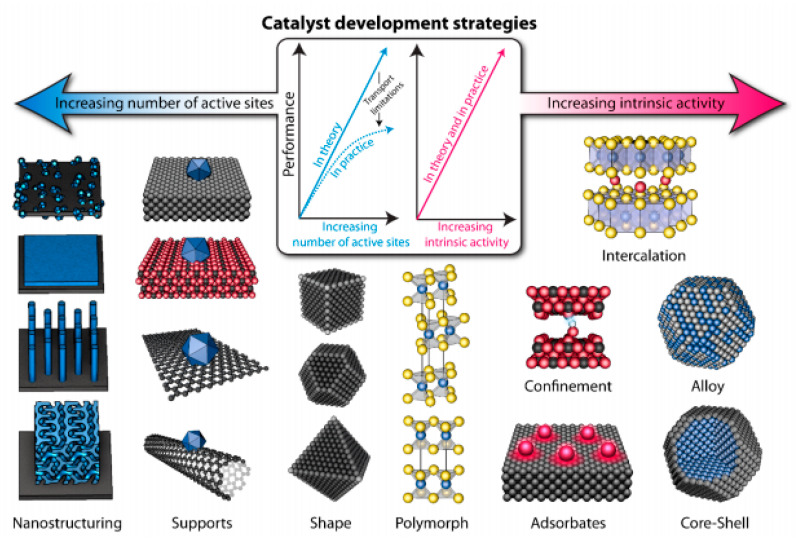
Catalyst development strategies. Schematic of various catalyst development strategies, which aim to increase the number of active sites and/or increase the intrinsic activity of each active site.

**Figure 10 molecules-25-04996-f010:**
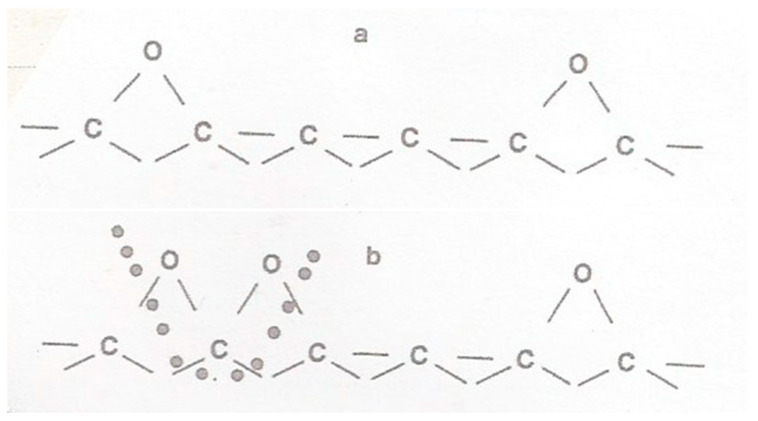
(**a**) Formation of C_2_O. (**b**) Formation of CO_2_.

**Figure 11 molecules-25-04996-f011:**
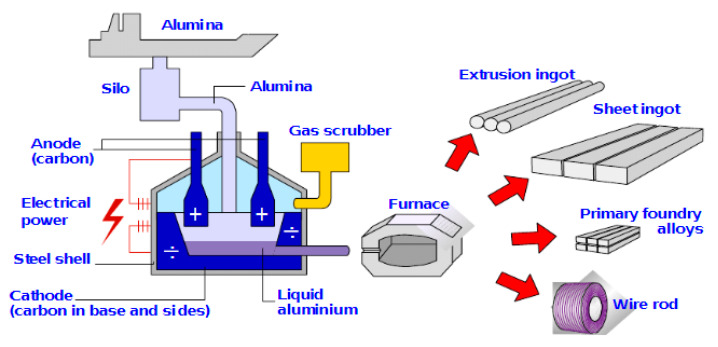
Scheme of aluminium production cell.

**Figure 12 molecules-25-04996-f012:**
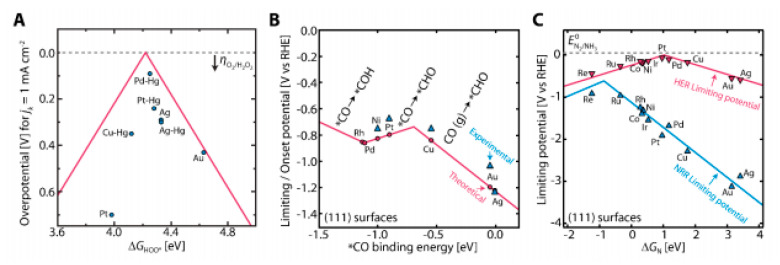
Emerging reactions of interest. (**A**) Volcano plot for hydrogen peroxide production on metals and alloys [254]. (**B**) Volcano plot for carbon dioxide reduction on metals [275,276]. (**C**) Volcano plot for nitrogen reduction on metals, with that of HER overlaid for comparison [286].

**Figure 13 molecules-25-04996-f013:**
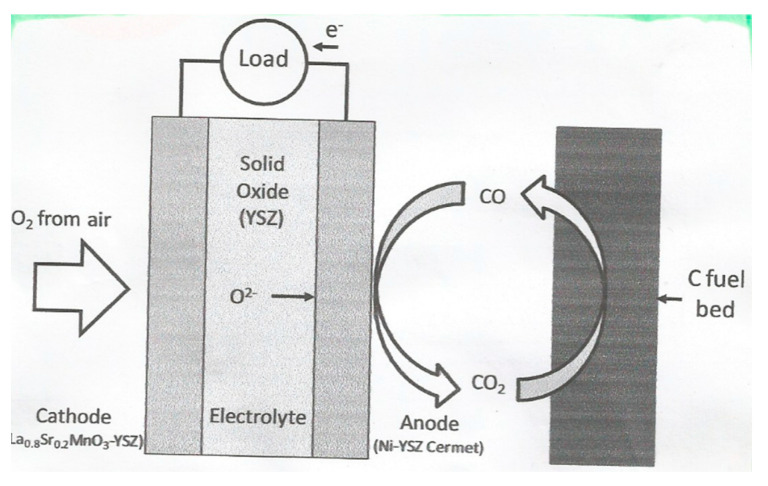
Operating schematic of a gasification-driven direct carbon solid oxide fuel cell (SOFC).

**Figure 14 molecules-25-04996-f014:**
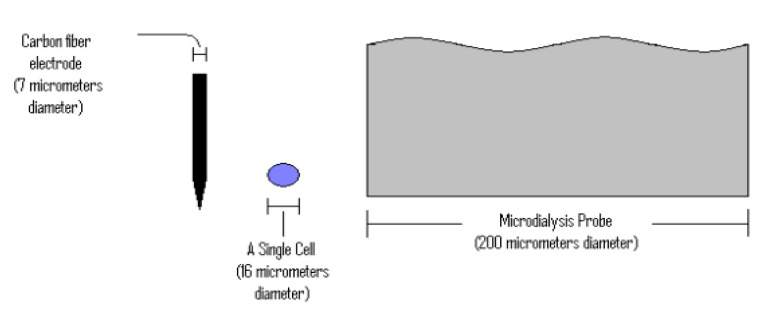
The relative sizes of a carbon fiber electrode and a microdialysis probe next to a single cell.

**Table 1 molecules-25-04996-t001:** Types of direct carbon fuel cells.

Fuel/Anode	Electrolyte	Cathode	T, °C
Solid graphite rod as fuel & anode C+4OH^−^ = 2H_2_O + CO_2_ + 4e^−^	Molten Hydroxides OH^−^	Air as oxidant O_2_ + 2H_2_O + 4e^−^ = 4OH^−^	600
Carbon particles as fuel in molten carbonate & anode C+2CO_3_^2−^ = 3CO_2_ + 4e^−^	Molten Carbonates CO_3_^2−^	Air as oxidant O_2_ + 2CO_2_+4e^−^ = 2CO_3_^2−^	800
Carbon particles in a fluidized bed C + 2O^2−^ = CO_2_ + 4e^−^	Oxygen ion conducting ceramic electrolyte O^2−^	Air as oxidant O_2_ + 4e^−^ = 2O^2−^	800 to 950
Fuel in contact with molten tin Sn + 2O^2−^ = SnO_2_ + 4e^−^			
Carbon particles as fuel in molten carbonate & anode C + 2O^2−^ = CO_2_ + 4e^−^			

**Table 2 molecules-25-04996-t002:** Tecnology status of various types of direct carbon fuel cells (DCFCs).

DCFC Technology	Status
Molten hydroxide	Average power densities of 40 mWcm^−2^ for over 540 h of operation with peak power density of 180 mWcm^−2^. The maximum efficiency achieved is 60%.
Molten carbonate	Power densities to 100–120 mWcm^−2^ and 80% efficiency.
-Solid Oxide -Solid carbon feed -Carbon mixed with molten metal -Carbon mixed with molten carbonate	The peak power density achieved is reported to be 140 mWcm^−2^ at 900 C. The peak power density achieved so far is about 160 mWcm^−2^ and 80 mWcm^−2^ respectively from hydrogen and liquid fuel JP-8. The peak power density achieved is 120 mWcm^−2^ using acetylene black as the fuel. SRI International has tested a 6 W 6-cell demonstration stack using different fuels.

**Table 3 molecules-25-04996-t003:** Characteristics of selected electrochemical energy storage technologies.

Characteristics	Capacitor	Supercapacitor	Battery
Specific energy (W h kg^−1^)	<0.1	1–10	10–100
Specific power (W kg^−1^)	>10.000	500–10,000	<1000
Discharge time	10^−6^ to 10^−3^	s to min	0.3–3 h
Charge time	10^−6^ to 10^−3^	s to min	1–5 h
Coulombic efficiency (%)	About 100	85–98	70–85
Cycle-life	Almost infinite	>500,000	About 1000

**Table 4 molecules-25-04996-t004:** Comparison between batteries and supercapacitors.

Comparison Parameter	Battery	Supercapacitor
Storage mechanism	Chemical	Physical
Power limitation	Reaction kinetics, mass transport	Electrolyte conductivity
Energy storage	High (bulk)	Limited (surface area)
Charge rate	Kinetically limited	High, same as discharge
Cycle life limitations	Mechanical stability, chemical reversibility	Side reactions

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
