# Peer review of "Carbon Anode in Carbon History"

_molecules, 2020, doi:10.3390/molecules25214996_

Round 1

Reviewer 1 Report

The author prepared a review manuscript entitled “Carbon anode in carbon history" This review manuscript will helpful for young researcher for developing high efficient carbon electrode materials for different energy storage devices, such as solar cells, fuel cells and supercapacitor. Hence, I suggested that this review manuscript will be suitable for publication in “Molecules” after address the following comments. 1. Authors should include some recent references in the revised manuscript. 2. Authors should carefully check the whole manuscript for typo errors and grammatical errors.

Reviewer 2 Report

This is an interesting and valuable review. Based on the historical perspective, the review also demonstrates stimulating perspective in near future.

This article is recommendable for the publication as it is.

Reviewer 3 Report

This review systematically describe the 10 development of various carbon materials with size, dopants, shape and structure designed to 11 achieve high catalytic electroactivity. In addition, they describe carbon applications in chemical 13 technology, neurochemical monitoring, electrode kinetics, direct carbon fuel cells, lithium ion 14 batteries, electrochemical capacitors, and supercapattery. In my opinion, it is a good review. But the title carbon anode in carbon history  is not suitable. I suggest author change the title and focus. In the manuscript, author introduce the history and carbon application. I do not think author focus on the one or two points. A good review should be has the core clue and then broaden the content. Finally, author do the great job but should focus on one point and deeply understand all the question about that to help author to systematically understand carbon.

Round 2

Reviewer 3 Report

no

This manuscript is a resubmission of an earlier submission. The following is a list of the peer review reports and author responses from that submission.